

# Relationships between the planetary boundary layer height and

# surface pollutants derived from lidar observations over China

7               Tianning Su[1], Zhanqing Li[1,2*], Ralph Kahn[3]

[1]Department of Atmospheric and Oceanic Sciences & ESSIC, University of Maryland, College Park, M

11                              aryland 20740, USA

[2]State Key Laboratory of Earth Surface Processes and Resource Ecology and College of Global Change

13              and Earth System Science, Beijing Normal University, 100875, Beijing, China

14       [3]Climate and Radiation Laboratory, Earth Science Division, NASA Goddard Space Flight Center,

15                          Greenbelt, MD, USA

* *Correspondence to:* Zhanqing Li (zli@atmos.umd.edu)



**Abstract.** The frequent occurrence of severe air pollution episodes in China has raised great concerns
with the public and scientific communities. Planetary boundary layer height (PBLH) is a key factor in
the vertical mixing and dilution of near-surface pollutants. However, the relationship between PBLH and
surface pollutants, especially particulate matter (PM) concentration, across the whole of China, is not yet
well understood. We investigate this issue at ~1500 surface stations using PBLH derived from space-
borne and ground-based lidar, and discuss the influence of topography and meteorological variables on
the PBLH-PM relationship. A generally negative correlation is observed between PM and the PBLH,
albeit varying greatly in magnitude with location and season. Correlations are much weaker over the
highlands than plains regions, which may be associated with lower pollution levels and mountain breezes.
The influence of horizontal transport on surface PM is considered as well, manifested as a negative
correlation between surface PM and wind speed over the whole nation. Strong wind with clean upwind
sources plays a dominant role in removing pollutants, and leads to weak PBLH-PM correlation. A
ventilation rate is introduced to jointly consider horizontal and vertical dispersion, which has the largest
impact on surface pollutant accumulation over the North China Plain. Aerosol absorption feedbacks also
appear to affect the PBLH-PM relationship, as revealed via comparing air pollution in Beijing and Hong
Kong. Absorbing aerosols in high concentrations likely contribute to the significant PBLH-PM
correlation over the North China Plain (e.g., during winter). As major precursor emissions for secondary
aerosols, sulfur dioxide, nitrogen dioxide, and carbon monoxide have similar negative responses to
increased PBLH, whereas ozone is positively correlated with PBLH over most regions, which may be
caused by heterogeneous reactions and photolysis rates.



## 1. Introduction


In the past few decades, China has been suffering from severe air pollution, caused by both
particulate matter (PM) and anthropogenic gases. PM pollutants are of greatest concern to the public
partly because they are much more visible (Chan and Yao, 2008; J. Li et al., 2016), and because they
have discernible adverse effects on human health. Moreover, airborne particles critically impact Earth's
climate through aerosol direct and indirect effects (Ackerman et al., 2004; Boucher et al., 2013; Guo et
al., 2017; Kiehl et al., 1993; Li et al., 2016; 2017a).
Multiple factors contribute to the severe air pollution over China. Strong emission due to rapid
urbanization and industrialization is a primary cause. Meanwhile, meteorological conditions and
diffusion within the planetary boundary layer (PBL) also play important roles in the exchange between
polluted and clean air. Among the meteorological parameters of importance, the PBL height (PBLH) can
be related to the vertical mixing, affecting the dilution of pollutants emitted near the ground through
various interactions and feedback mechanisms (Emeis and Schäfer. 2006; Seibert et al., 2010; Su et al.,
2017a). Therefore, PBLH is a critical parameter affecting near-surface air quality, and it serves as a key
input for chemistry transport models (Knote et al., 2015; LeMone et al., 2013). The PBLH can
significantly impact aerosol vertical structure, as the bulk of locally generated pollutants tends to be
concentrated within this layer. Turbulent mixing within the PBL can account for much of the variability
in near-surface air quality. On the other hand, aerosols can have important feedbacks on PBLH,
depending on the aerosol properties, especially their light absorption (e.g., black, organic, and brown
carbon; Wang et al., 2013). Multiple studies demonstrate that absorbing aerosols tend to affect surface
pollution in China through their interactions with PBL meteorology (Ding et al., 2016; Miao et al., 2016;
Dong et al., 2017; Petäjä et al., 2016). However, the importance and magnitude of aerosol feedback to



PBLH are still uncertain, as the feedback is closely related to aerosol structure, and may be weakened by
strong turbulence in PBL. Li et al. (2017b) give evidence of this in a review of the interaction between
the PBL and air pollution.

There are various methods for identifying the PBLH. The traditional and most common ones are

gradient (e.g., Johnson et al., 2001; Liu and Liang, 2010) and Richardson number methods (e.g.,
Vogelezang and Holtslag, 1996), both of which are typically based on temperature, pressure, humidity,
and wind speed profiles obtained by radiosondes. By using fine-resolution radiosonde observations, Guo
et al. (2016) obtained the first comprehensive PBLH climatology over China. Ground-based lidars, such
as the micropulse lidar (MPL), are also widely used to derive the PBLH (e.g., Hägeli et al., 2000; He et
al., 2008; Sawyer and Li, 2013; Tucker et al., 2009; Yang et al., 2013). The lidar-based PBLH
identification relies on the principle that a temperature inversion often exists at the top of the PBL,
trapping moisture and aerosols (Seibert et al., 2000), which causes a sharp decrease in the aerosol
backscatter signal at the PBL upper boundary. However, using ground-based observations to retrieve the
PBLH suffers from poor spatial coverage and very limited sampling. The Cloud-Aerosol Lidar with
Orthogonal Polarization (CALIOP) on board the Cloud-Aerosol Lidar and Infrared Pathfinder Satellite
Observations (CALIPSO) satellite (Winker et al., 2007), an operational spaceborne lidar, can retrieve
cloud and aerosol vertical distributions at moderate vertical resolution, complementing ground-based
PBLH measurements. Several studies already demonstrate both the effectiveness and the limitations of
using CALIPSO data for PBLH detection, showing sound but highly variable agreement with those from
radiosonde- and MPL-based PBLH results (Su et al., 2017b; Leventidou et al., 2013; Liu et al., 2015;
Zhang et al., 2016).

Several studies have explored the relationship between PBLH and surface pollutants in China. Tang



et al. (2016) used ceilometer measurements to derive long-term PBLH behavior in Beijing, further
demonstrating the strong correlation between the PBLH and surface visibility under high humidity
conditions. Wang et al. (2017) classified atmospheric diffusion conditions based on PBLH and wind
speed, and identified significant surface PM changes that also varied with dispersion conditions. Miao et
al. (2017) investigated the relationship between summertime PBLH and surface PM, and discussed the
impact of synoptic patterns on the development and structure of the PBL. Qu et al. (2017) derived one-
year PBLH variations from lidar in Nanjing, and presented the strong correlation between PBLH and
$PM_{2.5}$, especially for hazy and foggy days.

However, the majority of the studies mostly employ data only at a few stations. Yet, the interaction

between PBLH and surface pollutants under different topographic and meteorological conditions is not
well understood. Assessing the relationship between PM and the PBLH quantitatively over the entire
country, is of particular significance. PBL turbulence is not the only factor affecting air quality, so there
can be large regional differences in the interaction between the PBLH and PM. As such, the contributions
of various factors to the PBLH-PM relationship may be disclosed, that thus warrant a further investigation.

Given the above-mentioned limitations, this study presents a comprehensive exploration of the

relationship between the PBLH and surface pollutants over China, for a wide range of atmospheric,
aerosol and topographic conditions. Since 2012, China has drastically increased the number of
instruments and implemented rigorous quality control measures to measure hourly pollutant
concentrations nationally, of much better quality than previously available. The pollutant data derived
from surface observations, along with CALIPSO measurements, offer us an opportunity to investigate
the impact of PBLH on air quality on a nationwide basis. Regional characteristics and seasonal variations
are considered. Moreover, multiple factors related to the interaction between the PBLH and PM are



investigated, including surface topography, horizontal transport, and aerosol type. The relationships
between the PBLH and several gas pollutants are also presented. These empirical relationships between
PBLH and surface pollutants are aimed at improving our understanding and forecasting ability for air
pollution, as well as helping refine meteorological and atmospheric chemistry models.

**2.  Data and Method**
**2.1. Surface observation sites**

The topography of China is presented in Figure 1a, and the pink rectangles outline the four regions

of interest (ROI) for the current study: northeast China (NEC), the Yangtze River Delta (YRD), Pearl
River Delta (PRD), and North China Plain (NCP). The environmental monitoring station locations are
indicated with red dots in Figure 1b. They routinely measure hourly pollutant data, including PM with
diameters $\leq$ 2.5 and 10 $\mu$m ($PM_{2.5}$ and $PM_{10}$, respectively), sulfur dioxide ($SO_2$), nitrogen dioxide ($NO_2$),
carbon monoxide (CO), and ozone ($O_3$). The locations of meteorological stations operated by the China
Meteorological Administration are indicated in Figure 1c. We use wind speed and wind direction data
obtained at these stations. As shown in Figure 1d, blue lines represent the ground tracks over China for
the daytime overpasses of CALIPSO. To match the CALIPSO retrievals with surface pollutant and
meteorological data, we average the available CALIPSO retrievals within 35 km of the surface stations,
and use the noontime surface data, where "noontime" refers to results averaged from 1300 to 1500 China
standard time (CST). We also utilized the MPL data at Beijing and sun-photometer data at Beijing and
Hong Kong, two megacities located over NCP and PRD respectively. The MPL located at Beijing was
operated continuously by Peking University (39.99°N, 116.31°E) from Apr 2016 to Dec 2017, with a
temporal resolution of 15s and a vertical resolution of 15m. The near-surface blind zones for both lidars



are around 150 meters. Background subtraction, saturation, after-pulse, overlap, and range corrections
are applied to raw MPL data (He et al., 2008, Yang et al., 2013). In this study, Level 1.5 AOD at 550/440
nm and single-scattering albedo (SSA) at 675 nm at Beijing RADI (40°N, 116.38°E) and Hong Kong
PolyU (22.3°N, 114.18°E) Aerosol Robotic Network (AERONET) sites with hourly time resolution are
used.

**2.2. PBLH derived from MPL**

MPL data from Beijing were used to retrieve the PBLH for this study. Multiple methods have been

developed for retrieving the PBLH from MPL measurements, such as signal threshold (Melfi et al., 1985),
maximum of the signal variance (Hooper and Eloranta, 1986), minimum of the signal profile derivative
(Flamant et al., 1997), and wavelet transform (Cohn and Angevine, 2000; Davis et al., 2000). In this
study, we implement a well-established method by Yang et al. (2013) to derive the PBLH from MPL data,
with a few modifications. This method can handle all possible weather conditions and aerosol layer
structures, and is tested to be suitable for processing long-term lidar data. Initially, the first derivative of
a Gaussian filter with a wavelet dilation of 60 m is applied to smooth the vertical profile of MPL signals,
and to produce the gradient profile. The aerosol stratification structure is indicated by multiple valleys
and peaks in the gradient profile. To exclude misidentified elevated aerosol layers above the PBL, the
first significant peak in the gradient profile (if one exists) is considered the upper limit in searching for
the PBL top. Then, the height of the deepest valley in the gradient profile is attributed to the PBLH;
discontinuous or false results caused by clouds are subsequently eliminated manually. In this study, we
further estimated the shot noise (σ) induced by background light and dark current for each profile, and
then added threshold values of $\pm3\sigma$ to the identified peaks and valleys of this profile to reduce the



impact of noise. To validate MPL-derived PBLH, the values are compared with summertime radiosonde
PBLH data at 14:00 CST retrieved at Beijing station (39.80°N, 116.47°E) from potential temperature
profiles by the Richardson number methods (e.g., Vogelezang and Holtslag, 1996). Figure S1a shows
good agreement (R=0.7) between MPL- and radiosonde-derived PBLHs over Beijing.

**2.3. PBLH derived from CALIPSO**

CALIOP aboard the CALIPSO platform is the first space-borne lidar optimized for aerosol and cloud
profiling. As part of the Afternoon satellite constellation, or A-Train (L'Ecuyer and Jiang, 2010),
CALIPSO is in a 705-km Sun-synchronous polar orbit between 82°N and 82°S, with equator crossings
at approximately 1330 and 0130 local time and a 16-day repeat cycle (Winker et al., 2007, 2009).
CALIOP measures the total attenuated backscatter-coefficient (TAB) with a horizontal resolution of 1/3
km and a vertical resolution of 30 m in the low and middle troposphere, and has two channels (532 and
1064 nm). As the nighttime heavy surface inversion and residual layers tend to complicate the
identification of the PBLH, we only utilize daytime TAB data (Level 1B) in this study. For retrieving the
PBLH from CALIPSO, we typically use the maximum standard deviation (MSD) method, which was
first developed by Jordan et al. (2010) and then modified by Su et al. (2017b). In general, it determines
the PBLH as the lowest occurrence of a local maximum in the standard deviation of the backscatter
profile, collocated with a maximum in the backscatter itself. The PBLH retrieval range (0.3~4km),
surface noise check, and removal of attenuating and overlying clouds are subsequently included in this
method. In addition, due to the viewing geometry of the instrument, we define a constraint function:
$$\beta(i) = \max\{f(i+2), f(i+1)\} - \min\{f(i), f(i-1)\} \ , \qquad (1)$$



where $f(i+2)$, $f(i+1)$, $f(i)$, $f(i-1)$ are four adjacent altitude bins in the 532-nm TAB and where
the altitude decreases with increasing bin number i. To eliminate the local standard deviation maximum
caused by signal attenuation, we add the constraint $\beta > 0$, and locate the PBLH at the top of the aerosol
layer. Moreover, we also use the wavelet covariance transform (WCT) method to retrieve the PBLH, and
this retrieval serves as a constraint. We eliminate cases when the difference between the MSD and WCT
retrievals is above 0.5 km, to increase the reliability of the MSD retrievals.
Due to the high signal-to-noise ratio and reliability of MPL measurements, we use MPL-derived
PBLH to test the CALIPSO retrievals. The comparison between CALIPSO- and MPL-derived PBLH at
Beijing and Hong Kong (result from Su et al., 2017b) are shown in Figure S1b-c. Reasonable agreement
between CALIPSO- and MPL-derived PBLHs at these two sites is shown. The correlation coefficients
are above 0.6, which is similar to results from previous studies (e.g., Liu et al., 2015; Su et al., 2017b;
Zhang et al., 2016). Besides the differences in signal-to-noise ratio, the 10-40 km distance between the
MPL station and CALIPSO orbit also contributes to the differences between MPL- and CALIPSO-
derived PBLH.

**2.4. MODIS AOD data**
The MODIS instruments on board Terra and Aqua have 2330-km swath widths, and provide daily
AOD data with near-global coverage. In this study, we use Collection 6 MODIS level-2 AOD products
from the Aqua satellite at 550 nm (available at: https://www.nasa.gov/langley). AOD data are archived
with a nominal spatial resolution of 10 km × 10 km, and the data are averaged within 30 km radius
around the environmental stations to match with surface PM data. The MODIS land AOD accuracy is
reported to be $\pm(0.05+15\%$ AERONET AOD) (Levy et al., 2010).




## 3.  Results

### 3.1. Climatological patterns of PBLH and surface pollutants

The climatology of the PBLH, especially its seasonal variability, is very important for air-pollution-related studies. We utilized the CALIPSO measurements during the period 2006 through 2017 to represent the spatial distribution of seasonal mean PBLH, as shown in Figure 2a-d. A smoothing window of 20 km was applied to the original PBLH data at 1/3 km horizontal resolution. For comparison, we also used the PBLH data obtained from the Modern Era-Retrospective Reanalysis for Research and Applications (MERRA) reanalysis dataset with a spatial resolution of 2/3°×1/2° (longitude-latitude). The MERRA reanalysis data uses a new version of the Goddard Earth Observing System Data Assimilation System Version 5 (GEOS-5). The seasonal climatological patterns of MERRA-derived PBLH are presented in Figure 2e-h for the same period. In general, the climatological pattern of MERRA PBLH is similar to that of CALIPSO, though the MERRA values are higher in spring and summer, and the peak values are lower in autumn and winter. Both CALIPSO and MERRA PBLHs are generally shallower in winter, when the development of the PBL is typically suppressed by the weaker solar radiation reaching the surface, and is generally higher in summer, especially for inland regions.

Note that there are still large differences between CALIPSO- and MERRA-derived PBLH climatological patterns, which can be attributed to sampling biases, different definitions, and model uncertainty. First, since the spatial coverage and time resolution are quite different between the CALIPSO and MERRA datasets, the sampling used to calculate the climatologies are quite different. Moreover, MERRA PBLHs are derived from turbulent fluxes computed by the model, whereas CALIPSO usually identifies the top height of an aerosol-rich layer. Although turbulent fluxes would significantly affect





aerosol structures, the different definitions still can cause large differences between CALIPSO and
MERRA PBLHs. The detailed relationship between of CALIPSO- and MERRA PBLHs is presented in
Figure S1d. CALIPSO PBLH exhibits considerable differences from MERRA results, with a correlation
coefficients of ~0.4, indicating that the observations presented here will likely be useful for future model
refinement.

The seasonal mean values of CALIPSO and MERRA PBLHs over four ROIs are presented in Table

1. Broadly speaking, the differences between CALIPSO and MERRA PBLHs are much smaller than their
standard deviations. PBLH shows strong seasonality over NCP and NEC, ranging from ~0.9 km (winter)
to 1.5 km (summer). As the seasonal variation of PBLH is much smaller than the standard deviation over
PRD and YRD, the seasonal patterns are not clear for these two regions. MERRA PBLH shows similar
seasonal means with CALIPSO over NCP, with differences of ~0.1km, and shows the largest differences
(0.5km) with CALIPSO PBLH over NEC during winter.

Correspondingly, the seasonal means and standard deviations over four ROIs are listed in Table 1.

The $PM_{2.5}$ seasonal pattern is generally opposite that of PBLH, with the lowest values in summer and the
highest in winter. Since a high PBLH facilitates the vertical dilution and dissipation of air pollution, the
contrasting patterns of PBLH and $PM_{2.5}$ are consistent with expectation, although one cannot assure their
causal relationship from these plots alone. As this is a major polluted region, both PBLH and $PM_{2.5}$ show
particularly strong seasonality over NCP. PRD is a relatively clean region, and $PM_{2.5}$ maintains low values
($<50$ μg m$^{-3}$) through all seasons. The spatial distributions of $PM_{10}$, and multiple gas pollutants
($SO_2/NO_2/CO/O_3$) climatologies are shown in Figure S2. The seasonal and regional patterns of $PM_{2.5}$,
$PM_{10}$, $SO_2$, $NO_2$, and CO all show their highest values in winter and lowest in summer, similar to $PM_{2.5}$.
However, unlike the other pollutants, $O_3$ reaches its highest values during summer. These patterns are





discussed in more detail in subsequent sections.

**3.2. Regional relationships between PM and PBLH**

If the common factor driving large-scale variations in both PM and PBLH is meteorology, a regional

analysis of their relationship could elucidate the meteorological impacts. We investigate the CALIPSO-
PBLH and surface $PM_{2.5}$ data case by case. The scatterplots for annually aggregated PBLH versus surface
$PM_{2.5}$ for the four ROI are shown in Figure 3. Although there is a large spread and regional differences,
the negative correlations between PBLH and $PM_{2.5}$ are seen in all ROIs. PBLH values show the most
negative correlation with $PM_{2.5}$ over the NCP, with a correlation coefficient of -0.36. PBLH also shows
significant negative correlation with $PM_{2.5}$ over YRD and NEC, with correlation coefficients of -0.24
and -0.15, respectively. (Hereafter, "significant" indicates the correlation is statistically significant at the
95% confidence level.) The weak PBLH correlation with $PM_{2.5}$ over the PRD is not statistically
significant. The relationships between PBLH and $PM_{10}$ are similar to those with $PM_{2.5}$, except with larger
spreads, because the magnitudes of $PM_{10}$ are larger than those of $PM_{2.5}$ (Figure S3). Compared to
CALIPSO data, the MPL has a much higher signal-to-noise ratio and can continuously observe at one
location. Therefore, we compare the relationships between MPL-derived PBLH and $PM_{2.5}$ with those
from CALIPSO at Beijing (Figure S4). Similar to the relationship derived from CALIPSO, the PBLH
shows a significantly nonlinear relationship with $PM_{2.5}$ over Beijing (a major city in the NCP).

We notice that the ranges of $PM_{2.5}$ for these ROIs are significantly different; therefore, the

background pollution level is likely to be an important factor for the PBLH-PM relationship. We also
normalize the $PM_{2.5}$ by MODIS AOD, a widely used parameter to represent the columnar aerosol amount,
to qualitatively account for background or transported aerosol that is not concentrated in the PBL. The





relationship between PBLH and PM$_{2.5}$/AOD over four ROIs are presented in Figure 4. Clearly, after
normalizing PM$_{2.5}$ by AOD, the spread of these scatter plots and the regional differences are significantly
reduced, and the correlations became more significant for all ROIs, especially for PRD. This is because
transported aerosol aloft can contribute to variability in total column AOD that is unrelated to the PBLH.
Figure S5 provides a closer look at the regional differences among individual sites. As with Figure
3, the most negative correlations between PBLH and PM$_{2.5}$ appear over the NCP, likely a testament to
intense PBL-aerosol interactions, which may be caused by concentrated local sources. Several scattered
sites show positive correlations between PBLH and PM$_{2.5}$, though they are generally not significant. Note
that the PBLH-PM$_{2.5}$ correlations are apparently stronger for heavily polluted regions, than for clean
regions. However, after normalizing PM$_{2.5}$ by AOD, the correlations are improved preferentially for clean
regions (where aerosol aloft makes a larger fractional contribution to the AOD), and thus, the differences
between clean and polluted regions are reduced (Figure S6). It further indicates that the background
pollution level plays a critical role in the PBLH-PM relationship.
As the NCP experiences the most pronounced seasonality in both PBLH and PM$_{2.5}$, their
relationship over this region also shows the most prominent seasonal differences (Figure S5c-f). Figure
5 focuses on the seasonal dependence of the PBLH and PM$_{2.5}$ relationship over the NCP. The mean slope
for this region is ~90 μg m$^{-3}$ km$^{-1}$ during winter, and is only ~20 μg m$^{-3}$ km$^{-1}$ in summer. For comparison,
the annual aggregated relationship between PBLH and PM$_{2.5}$ is presented in Figure 5e. PM$_{2.5}$
concentrations do not increase linearly with decreasing PBLH. Specifically, PM$_{2.5}$ increases rapidly with
decreasing PBLH when PBLH is lower than 1 km, but changes much more slowly for PBLH > 1.5 km.
The seasonal mean values for PM$_{2.5}$ and PBLH are presented as colored dots in Figure 5e, and the
whiskers represent the standard deviations. For winter, the PBLH is generally shallow, PM$_{2.5}$



concentrations are high, and thus PBLH shows the most significant negative correlation with $PM_{2.5}$.
Conversely, in summer, the PBLH is generally higher, $PM_{2.5}$ concentrations are lower, and the PBLH-
$PM_{2.5}$ relationship is virtually flat. Such seasonally distinct PBLH-$PM_{2.5}$ relationships have not previously
been studied quantitatively, and can contribute to improving $PM_{2.5}$ predictions.

**3.3. Association with horizontal transport**

The PBLH mainly affects the vertical mixing and dispersion of air pollution, but horizontal transport

also plays a critical role in surface air quality. Figure 6a-b present the PBLH-$PM_{2.5}$ relationships over
China under strong wind (WS>4m s$^{-1}$) and weak wind (WS<4m s$^{-1}$) conditions. In addition, Figure 6c-d
show the aerosol extinction profiles as a function of PBLH under strong and weak wind conditions. The
aerosols extinction coefficients are retrieved by the MPLs at Beijing, and the Klett method is applied
(Klett, 1985). Under strong wind conditions, $PM_{2.5}$ is much less sensitive to PBLH than for weak wind.
In both strong and weak wind conditions, aerosol structure changes systematically with PBLH, and sharp
aerosol extinction gradients appear at the top of the PBL. Nonetheless, under strong wind, the aerosol
extinction is typically low in the PBL, and the surface extinction is even lower than the extinction at PBL
top. In this situation, the strong wind likely plays a dominant role in affecting $PM_{2.5}$ concentration. Under
weak wind, the response of near-surface pollutants to PBLH is highly nonlinear, and both aerosol
extinction and $PM_{2.5}$ fall rapidly as the PBLH increases from 600m to 1200m.

We further consider the relationship between PBLH-$PM_{2.5}$ under different wind-direction regimes

for Beijing. Two different regimes are easy to identify: a northerly wind and a southerly wind; these are
divided by the red line in Figure 7a. The northerly air comes from arid and semiarid regions in northwest
China and Mongolia, and is usually strong and clean. The southerly wind comes from the southern part



of the NCP, with high humidity and aerosol content. To relate the connections between WS, PBLH, and
surface air quality, at least qualitatively, we define the ventilation rate (VR) as VR = WS × PBLH (Tie et
al., 2015). Figures 7b-e present the PBLH-PM$_{2.5}$ and VR-PM$_{2.5}$ relationships under southerly wind and
northerly wind conditions, respectively. For all wind conditions, VR shows reciprocal relationship with
surface PM$_{2.5}$. Under northerly wind conditions, both PBLH-PM$_{2.5}$ and VR-PM$_{2.5}$ relationships are flatter
and have lower correlation coefficients. The northerly wind is apparently effective in removing pollutants
and may play a dominant role in affecting air quality. For the southerly wind, the PM$_{2.5}$ concentration is
highly sensitive to PBLH and VR values.

To further illustrate the coupling effects of PBLH and WS on surface pollutants, Figure 8a presents

the relationship between noontime WS and PM$_{2.5}$ concentration across China. Overall, WS is negatively
correlated with PM$_{2.5}$, although a few stations over southwest China show positive correlations. A
negative correlation might be expected in general, as strong winds can be effective at removing air
pollutants; however, other factors such as wind direction must also be considered, as, for example,
upwind sources could increase pollution under higher wind conditions. There are positive correlations
between PBLH and near-surface WS in most cases (Figure S7a), and thus, low PBLH and weak WS tend
to occur together over much of China. These unfavorable meteorological conditions for air quality would
exacerbate severe pollution episodes.

To consider horizontal and vertical dispersion jointly, we investigate the nationwide relationships

between VR and PM$_{2.5}$. In general, VR is overwhelmingly negative correlated with surface PM$_{2.5}$ (Figure
S7b). Based on Figure 8a, VR is typically reciprocal to PM$_{2.5}$ for all wind conditions, and thus, we use
the function $f(x) = A/x$ to characterize the relationship between VR and PM$_{2.5}$, with A as the fitting
parameter, and $x$ is VR, and $f(x)$ is PM$_{2.5}$. The spatial distribution of A, presented in Figure 8b, shows the





331 largest values over the NCP, indicating that the $PM_{2.5}$ concentration is highly sensitive to the VR there.

332 Moreover, VRs are relatively large over the coastal areas, where sea-land breezes could play a role in

333 dispersing air pollution. The detailed relationships and fitting functions for four ROIs are presented in

334 Figure S8. We note that although there are large regional differences in the PBLH-$PM_{2.5}$ relationship

335 (Figure 3), the VR-$PM_{2.5}$ relationships are similar for the different study regions. Therefore, by combining

336 vertical and horizontal dispersion conditions, the overall VR apparently has a similar effect on $PM_{2.5}$ for

337 all four ROI.

339 **3.4. Correlations with topography**

340  The PBL structure and $PM_{2.5}$ concentration can both be affected by topography. We also divided all

341 the sites into two categories based on elevation: plains (elevation < 0.5 km) and highland (elevation > 1

342 km). Figure 9a-d presents the correlation coefficients and slopes between $PM_{2.5}$ and PBLH for the plains

343 and highland areas. Much stronger correlations exist in the plains than the highlands. A reciprocal

344 correlation is shown between station elevation and the PBLH-$PM_{2.5}$ slope (Figure 9e). The magnitudes

345 of slopes decrease dramatically with elevation increase between 0 and 500 m. Local emissions also affect

346 aerosol loading, and differences between plains and highland areas regarding local source activity could

347 be important here as well. Figure 9e shows that the low-elevation regions are typically more polluted

348 than highland areas, and the magnitudes of PBLH-$PM_{2.5}$ slopes also tend to be higher.

349  Returning to Figure S5, much stronger correlations for PBLH-$PM_{2.5}$ relationships are found over

350 polluted regions, which also correspond to the plains areas due to strong local emissions. Therefore, high

351 aerosol loading is likely to be another factor contributing to the strong correlation between PBLH and

352 $PM_{2.5}$ over the plains, whereas the low $PM_{2.5}$ concentration may contribute to the weak PBLH- $PM_{2.5}$



correlation over the highlands.
In addition, horizontal transport is associated with topography. Thus, we illustrate the distribution
of WS for plains and highland areas in Figure 9f. Clearly, WS is generally larger for highland areas,
especially for strong wind cases. In fact, the 10% and 25% quantiles of WS are nearly the same between
plains and highland areas, whereas there are apparent differences in the 75% and 90% quantiles. Strong
wind cases account for 37% of the total over highland areas, and only account for 27% of the total over
the plains. As discussed in section 3.3, strong wind can effectively remove surface pollutants, and can
play a dominant role in affecting pollution levels. In this situation, PBLH might not play as critical a role
in PM concentration. Thus, mountain slope winds, along with less local emission, are likely to be leading
factors accounting for the differences in PBLH-$PM_{2.5}$ correlations between plains and highland areas.

**3.5. Correlations between gaseous pollutants and PBLH**
Secondary aerosol contributes significantly to the surface PM concentration over China (Huang et
al., 2014). Multiple gas pollutants, such as $SO_2$, $NO_2$, and CO, are major precursor emissions for the
formation of secondary aerosols, which are closely related to $PM_{2.5}$ concentration (Guo et al., 2014; Wang
et al., 2016). Further, the near-surface concentrations of these gaseous pollutants can also have severely
negative effects on the environment and human health. We investigate the relationships between gaseous
pollutants and the PBLH due to their importance, by matching the CALIPSO PBLH with
$SO_2$/$NO_2$/CO/$O_3$ concentrations obtained from surface stations (Figure 10). Again, the relationships
between CALIPSO PBLH and $SO_2$/$NO_2$/CO/$O_3$ are similar to those derived from MPLs (Figure S4). For
$SO_2$, $NO_2$, and CO, the correlations with PBLH are similar to the PBLH-PM correlations over NCP, but
slightly weaker.





Similar to PBLH-PM relationships, the correlations between PBLH and $SO_2$/$NO_2$/CO are negative

for all ROIs. This is understandable, because the PBLH is likely to play a role in the vertical dilution and

dissipation of most gaseous pollutants. However, $O_3$ shows a positive correlation with PBLH for all ROIs,

which might be due to $O_3$ photochemistry. As radiation reaching the surface increases, convection is

enhanced and the PBLH tends to grow higher. At the same time, increased insolation with sufficient

precursor emissions ($NO_x$, CO, and $VOC_s$) can increase the net photochemical production of $O_3$.

Therefore, higher $O_3$ concentrations and high PBLH could occur together.    Moreover, when the PBL is

shallow and aerosol concentration is high, heterogeneous reactions on surfaces of multiple aerosols (e.g.

sulfate, mineral dust, and organic carbon aerosols) can uptake ozone precursors such as $NO_x$ and $N_2O_5$,

and thus, reduce the ozone production (Ravishankara, 1997; Jacob., 2000). And Liao and Seinfeld (2005)

found that the high aerosol loading reduces ozone concentrations by 25-30% through heterogeneous

reactions over eastern China. Taken together, decreased PBLH correlates with increased near-surface

aerosol concentration, leading to a reduction in precursors required for $O_3$ production, and an increase in

$O_3$ destruction by heterogeneous reactions. This could explain, at least qualitatively, the positive PBLH-

$O_3$ relationship.

**3.6. Potential feedback of absorbing aerosols**

Depending on their radiative properties, aerosols can have feedbacks on the PBLH. Multiple studies

point out a positive feedback between absorbing aerosols and the PBLH (Ding et al., 2016; Miao et al.,

2016; Petäjä et al., 2016). Using lidars and AERONET data, we examine the link between the PBLH-

$PM_{2.5}$ relationship and particle optical properties over Beijing and Hong Kong. We utilized AERONET

SSA data to classify aerosols as absorbing (SSA $\leq$ 0.85) or weakly absorbing (SSA > 0.9). The





correlation between PBLH and PM$_{2.5}$ is much stronger for absorbing cases over both Beijing and Hong
Kong (Figure 11). Noted the PBLHs over Beijing are obtained from MPL. Due to lack of available MPL
data, the PBLHs over Hong Kong are calculated by CALIPSO. Since AERONET SSA is more reliable
for the cases when AOD at 440nm is above 0.4 (Schafer et al., 2014), Figure S9 shows the PBLH-PM$_{2.5}$
relationship for absorbing and weakly absorbing cases over Beijing with a constraint of AOD$_{440}$>0.4. The
PBLH-PM$_{2.5}$ correlation remains considerably stronger for absorbing than weakly absorbing cases. Under
sufficient aerosol loading, we found PBLH-PM$_{2.5}$ correlations become stronger for both absorbing and
weakly absorbing cases. In addition, there are many more strongly absorbing cases for Beijing (~35%)
than for Hong Kong (~10%), and the total PBLH-PM$_{2.5}$ correlation is much stronger over Beijing.
Moreover, we show how absorbing optical depths over Beijing and Hong Kong correlate with the
general PBLH-PM$_{2.5}$ relationship in Figure 11e-f. Under highly absorbing optical depth conditions, PM$_{2.5}$
tends to be higher for a given PBLH. Large absorbing optical depths in Beijing offer great potential for
reducing the radiation reaching the surface, likely reducing the PBLH, and at the same time, heating the
middle and upper PBL, which would tend to cause a temperature inversion and increase the stability in
the PBL. The strongly absorbing aerosols with high loading are likely to give important feedback to
PBLH, and may contribute to the strong correlation between the PBLH and PM over Beijing. Other
factors could be involved, such as the vertical distribution of aerosol, the insolation, and the actual SSA
of the particles; further examination of these phenomena is beyond the scope of the current paper.

**4. Discussion and conclusions**
Based on ten years of CALIPSO measurements and other environmental data obtained from more
than 1500 stations, large-scale relationships between PBLH and PM are assessed over China. We observe



widespread negative correlations, albeit varying greatly in magnitude and seasonal timing by region.
Nonlinear responses of $PM_{2.5}$ to PBLH evolution are found, especially for NCP, the most polluted region
of China. Strongest PBLH-$PM_{2.5}$ interaction is found when the PBLH is shallow and $PM_{2.5}$ concentration
is high, which typically corresponds to the wintertime cases. Specifically, the negative correlation
between PBLH and $PM_{2.5}$ is most significant during winter. Moreover, we find that regional differences
in the PBLH-PM relationships are correlated with topography. Strong correlations between PBLHs and
aerosols occur in low-altitude regions. This might be related to the more frequent air stagnation and
strong local emission over China's plains, as well as a greater concentration of emission sources. The
mountain breezes and a larger fraction of transported aerosol above the PBL help weaken the PBLH-PM
correlation over highland areas.

As pollution levels can affect the PBLH-$PM_{2.5}$ relationship, we normalized $PM_{2.5}$ by MODIS total-

column AOD to account for the background aerosol in different regions. Comparing to PBLH-$PM_{2.5}$
correlations, the correlations between PBLH and normalized $PM_{2.5}$ ($PM_{2.5}$/AOD) increased significantly
for clean regions, resulting in smaller regional differences overall. Retrieving surface $PM_{2.5}$ from AOD
constraints has been investigated in many studies. The detailed relationships between PBLH and
$PM_{2.5}$/AOD over different ROIs are also expected to be significant for relating $PM_{2.5}$ to remotely sensed
AOD, due to the way PBLH affects near-surface aerosol concentration.

Horizontal transport also shows significant inverse correlation with $PM_{2.5}$ concentrations. WS and

PBLH tend to be positively correlated with each other in the study regions, which means meteorologically
favorable horizontal and vertical dispersion conditions are likely to occur together. Wind direction can
also significantly affect the PBLH-PM relationship. Strong wind with clean upwind sources plays a
dominant role in improving air quality, and leads to weak PBLH-PM correlation. The combination of



WS and PBLH, representing a "ventilation rate," shows a reciprocal correlation with surface PM in all
the regions studied. VR also is found to have the largest impact on surface pollutant accumulation over
the NCP.

As major precursor emissions for secondary aerosols, $SO_2$, $NO_2$, and CO show negative correlations

with PBLH, similar to the PBLH-PM correlations. However, $O_3$ is positively correlated with PBLH over
most regions, which may be caused by heterogeneous reactions and photolysis rates. This observation
merits further investigation using comprehensive measurements of chemical properties together with
necessary simulations from atmospheric chemistry model to ascertain the causes of the positive PBLH-
$O_3$ correlations.

As revealed by observations at Beijing and Hong Kong, absorbing aerosols with sufficient aerosol

loading likely contribute the strong PBLH-PM correlation. Large absorbing AOD would reduce the
radiation reaching the surface and heat the middle and upper PBL, which could increase the stability in
the PBL, representing a direct interaction between PBLH and PM. Much more strongly absorbing cases
for NCP than for PRD appear to contribute to the large contrast for PBLH-PM correlations between these
two regions. On the other hand, despite the strong correlations for absorbing cases with sufficient aerosol
loading, identifying a causal relationship between them is still elusive, as confounding factors, such as
aerosol vertical distribution, aerosol microphysical properties, ambient insolation, and meteorological
conditions, could all be involved. This merits further analysis using more comprehensive measurements
from field experiments, from which integrated aerosol conditions and model simulations can account for
aerosol radiative forcing while controlling all the other relevant variables

Our work comprehensively covers the relationships between PBLH and surface pollutants over



larger spatial scales in China. Multiple factors, such as horizontal transport, topography, and aerosol
optical properties, are found to be highly correlated with PBLH and near-surface aerosol concentration.
Such information can help improve our understanding for the complex interactions between air pollution
and meteorological factors, as well as help refine meteorological and atmospheric chemistry models. In
the future, we plan to combine field observation and numerical modeling for a more comprehensive,
quantitative study of the interaction between aerosol, wind, and PBL under different weather regimes
and geographic locations, in order to more fully characterize the nature of their interaction in the
atmosphere.

*Data availability.* The meteorological data are provided by the data center of China Meteorological
Administration (data link: http://data.cma.cn/en). The hourly pollutant data are released by the Ministry
of Environmental Protection of China (data link: http://113.108.142.147:20035/emcpublish). The
CALIPSO and MODIS data are obtained from the NASA Langley Research Center Atmospheric Science
Data Center (data link: https://www.nasa.gov/langley). The MERRA reanalysis data are publicly
available at https://disc.sci.gsfc.nasa.gov/datasets?page=1&keywords=merra. The AERONET data are
publicly available at https://aeronet.gsfc.nasa.gov.
*Competing interests.* The authors declare that they have no conflict of interest.

*Acknowledgements.* This work is supported in part by grants from the National Science Foundation (NSF)
(AGS1534670) and NSF of China (91544217) and NSF of US (). The authors would like to acknowledge
the Department of Atmospheric and Oceanic Sciences of Peking University for providing the ground-
based lidar data. We thank the Prof. Chengcai Li and Prof. Jing Li for theirs effort in establishing and
maintaining the MPL site. We greatly appreciate the helpful advice from Prof. Jing Li and Prof. Chengcai
Li at Peking University. We also thank the provision of surface pollutant data and meteorological data by
the Ministry of Environmental Protection and China Meteorological Administration. We extend sincerest
thanks to the CALIPSO, MODIS, AERONET, and MERRA teams for their datasets. The contributions
of R. Kahn are supported in part by NASA's Climate and Radiation Research and Analysis Program
under H. Maring, NASA's Atmospheric Composition Modeling and Analysis Program under R. Eckman.





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



**Table 1.**
Mean values and standard deviation (STD) of CALIPSO-PBLH, MERRA-PBLH, and $PM_{2.5}$ over
different ROIs.

| Parameter | | | NCP | PRD | YRD | NEC |
|---|---|---|---|---|---|---|
| CALIPSO-PBLH (km) | MAM | Mean | 1.40 | 1.35 | 1.31 | 1.40 |
| | | STD | 0.54 | 0.47 | 0.48 | 0.59 |
| | JJA | Mean | 1.47 | 1.27 | 1.24 | 1.46 |
| | | STD | 0.51 | 0.44 | 0.46 | 0.55 |
| | SON | Mean | 1.21 | 1.24 | 1.26 | 1.15 |
| | | STD | 0.45 | 0.36 | 0.39 | 0.50 |
| | DJF | Mean | 1.06 | 1.07 | 1.12 | 0.94 |
| | | STD | 0.40 | 0.34 | 0.41 | 0.47 |
| MERRA-PBLH (km) | MAM | Mean | 1.57 | 1.16 | 1.24 | 1.45 |
| | | STD | 0.75 | 0.53 | 0.47 | 0.69 |
| | JJA | Mean | 1.46 | 0.99 | 1.07 | 1.49 |
| | | STD | 0.72 | 0.36 | 0.39 | 0.68 |
| | SON | Mean | 1.37 | 1.18 | 1.22 | 1.19 |
| | | STD | 0.48 | 0.37 | 0.33 | 0.54 |
| | DJF | Mean | 1.08 | 1.09 | 1.05 | 0.65 |
| | | STD | 0.36 | 0.40 | 0.32 | 0.36 |
| $PM_{2.5}$ ($\mu g\ m^{-3}$) | MAM | Mean | 63.1 | 32.8 | 50.4 | 34.8 |
| | | STD | 45.1 | 22.1 | 29.2 | 29.4 |
| | JJA | Mean | 51.2 | 25.1 | 37.9 | 29.6 |
| | | STD | 36.8 | 20.4 | 24.1 | 24.4 |
| | SON | Mean | 70.9 | 39.3 | 42.4 | 44.2 |
| | | STD | 58.4 | 23.1 | 28.3 | 49.1 |
| | DJF | Mean | 102.7 | 44.2 | 69.8 | 60.3 |
| | | STD | 84.2 | 28.3 | 51.3 | 54.4 |





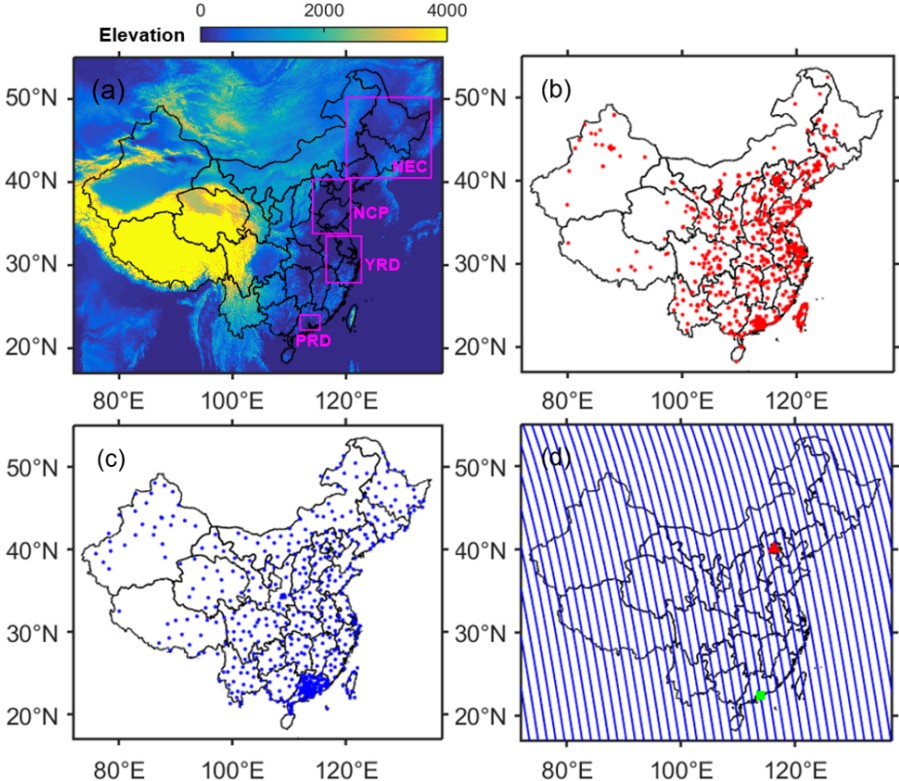

**Figure 1.** (a) Topography of China. The black rectangles outline the five regions of interest: northeast

China (NEC): 40.5-50.2°N, 120.1-135°E; North China Plain (NCP): 33.8-40.3°N, 114.1-120.8°E; Pearl

River Delta (PRD): 22.2-24°N, 111.9-115.4°E; and Yangtze River Delta (YRD): 27.9-33.5°N, 116.5-

122.7°E. Locations of (b) environmental stations and (c) meteorological stations. (d) Blue lines indicate

CALIOP daytime orbits (in ascending node). Ground-based lidar and sun-photometer are deployed at

Beijing (red circle), and sun-photometer is deployed at Hong Kong (green circle).

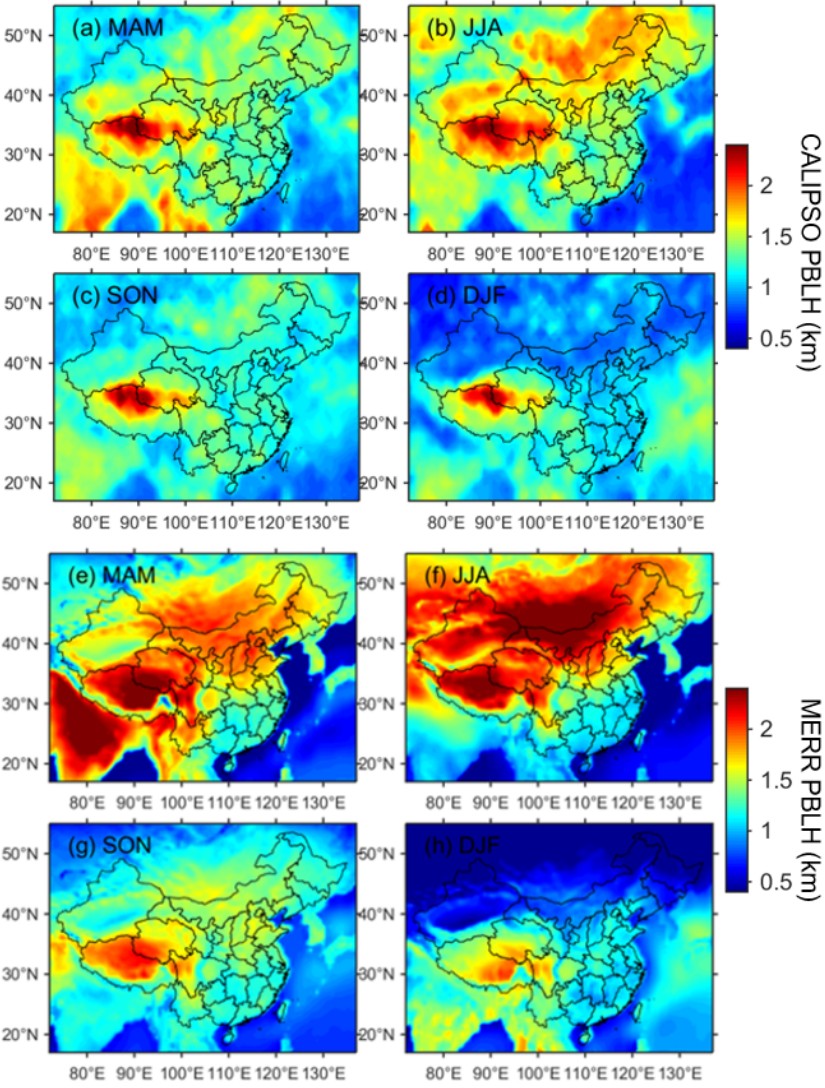

**Figure 2.** Spatial distributions of climatological mean PBLH derived from CALIPSO for (a) March-

April-May (MAM), (b) June-July-August (JJA), (c) September-October-November (SON), and (d)

December-January-February (DJF) during the period 2006–2017. Spatial distributions of climatological

mean noontime PBLH obtained from MERRA for (e) MAM, (f) JJA, (g) SON, and (h) DJF during the

same period.

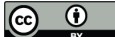



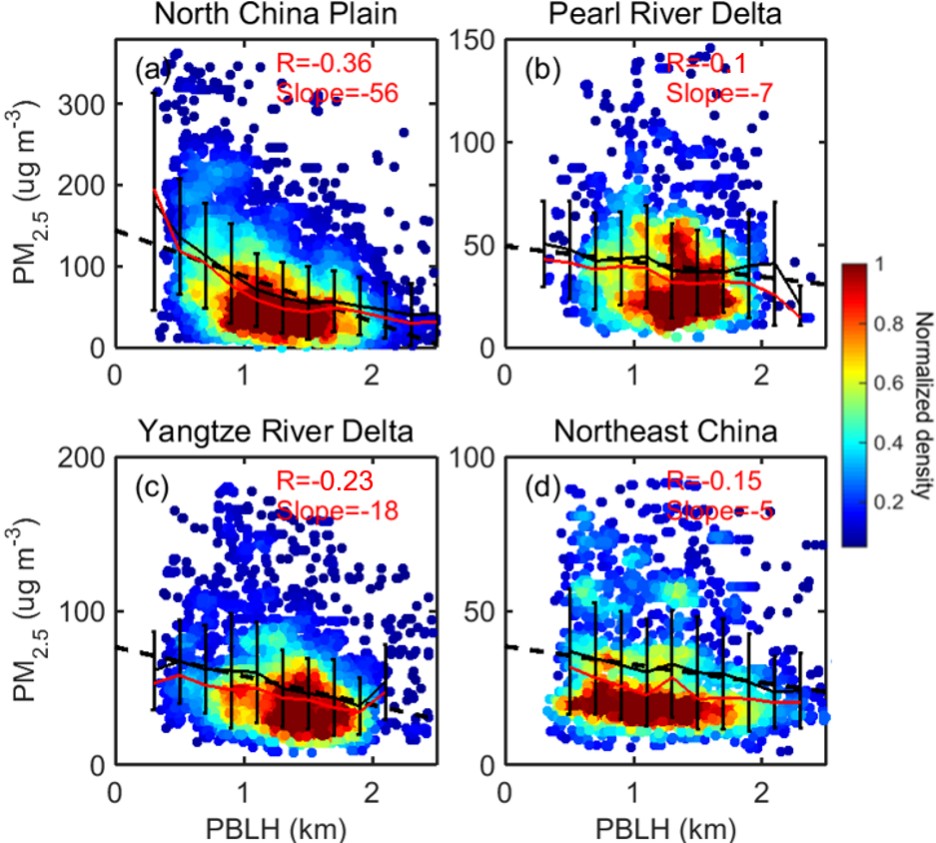

684

**Figure 3.** The relationship between CALIPSO-derived PBLH and noontime $PM_{2.5}$ over (a) NCP, (b) PRD,

(c) YRD, and (d) NEC. The black solid lines represent the average values for each bin, and whiskers

indicate one standard deviation. The red solid lines highlight the median for each bin, and the black

dashed lines give the linear regressions. The correlation coefficients and slopes for these relationships are

shown at the top of each panel. The color-shaded dots indicate the normalized sample density.





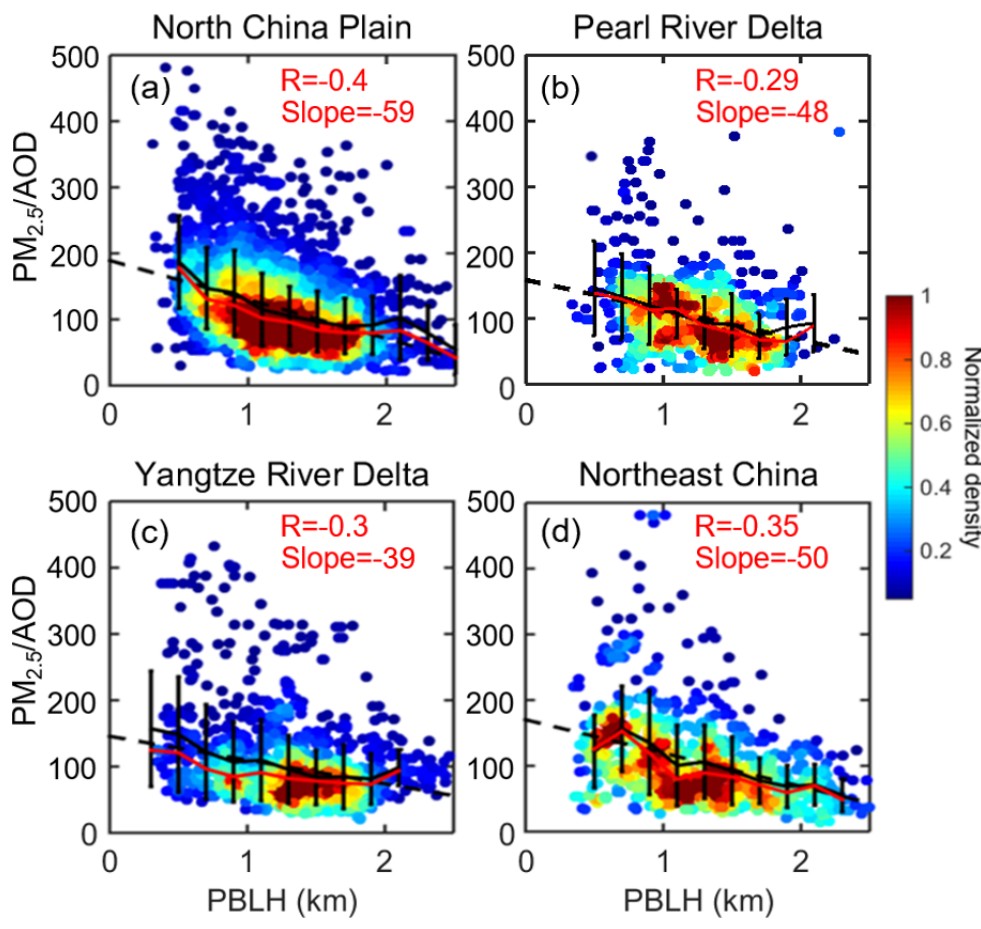


**Figure 4.** Similar to Figure 3, but for the relationship between CALIPSO PBLH and noontime

$PM_{2.5}$/AOD (unit: μg m$^{-3}$ per AOD) over four ROIs.





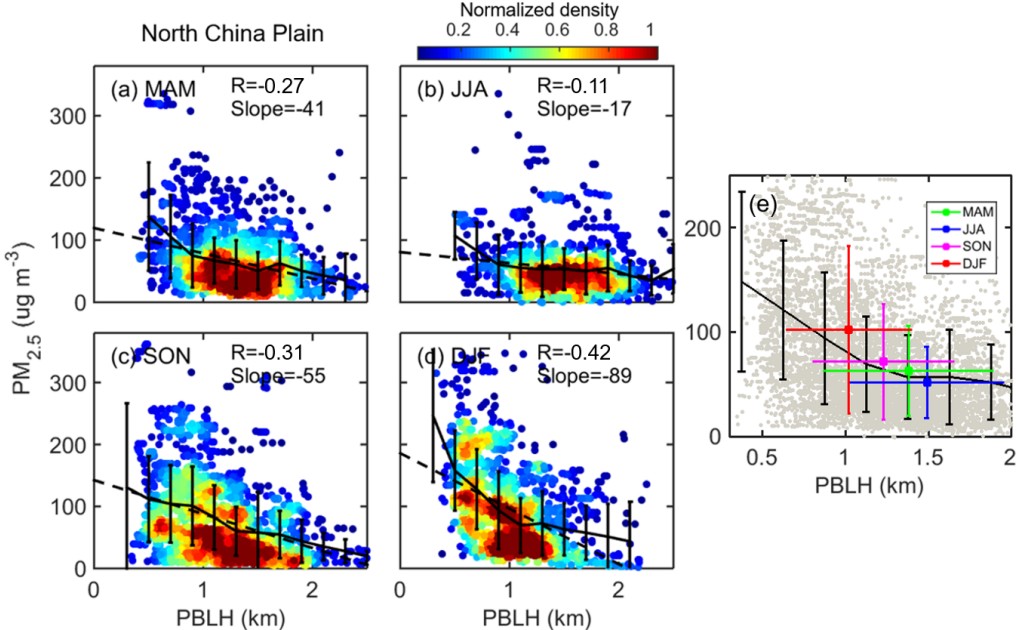

**Figure 5.** The relationship between CALIPSO PBLH and PM$_{2.5}$ over the NCP for (a) MAM, (b) JJA, (c)

SON, and (d) DJF. (e) General relationship between PM$_{2.5}$ and PBLH aggregated over all seasons, with

individual observations for each day plotted as gray dots. The green, blue, pink, and red dots present the

mean values for MAM, JJA, SON, and DJF, respectively.



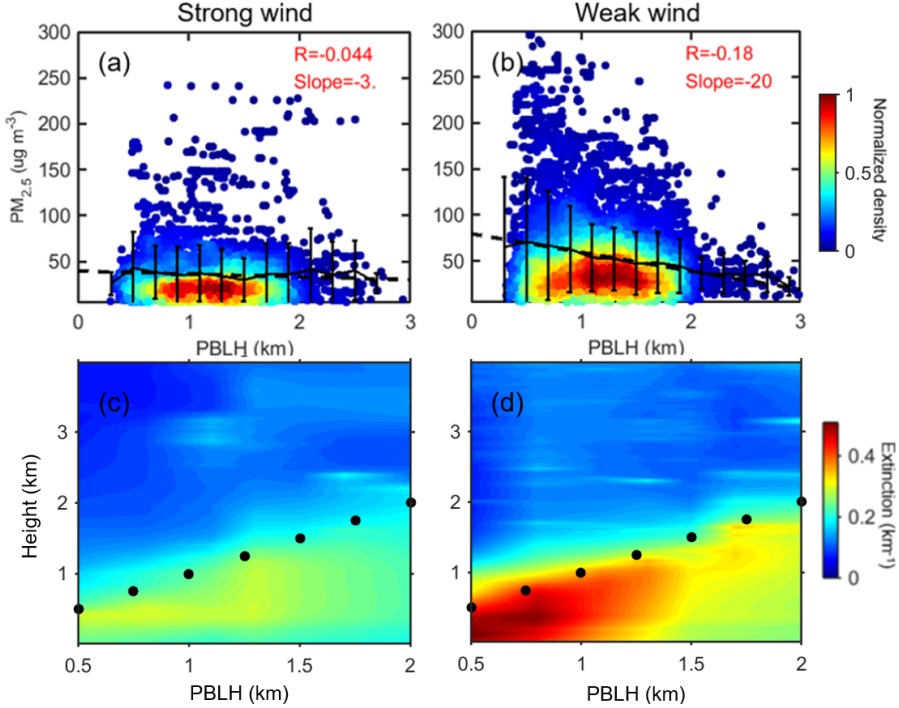


**Figure 6.** The relationship between CALIPSO PBLH and PM$_{2.5}$ over China for (a) strong wind (WS>4m

s$^{-1}$) and (b) weak wind (WS<4m s$^{-1}$). The aerosol extinction profiles at ~550 nm derived from MPL at
Beijing change with different MPL-derived PBLH under (c) strong wind and (d) weak wind conditions.
In (c, d), the black dots indicate the location of PBL top.





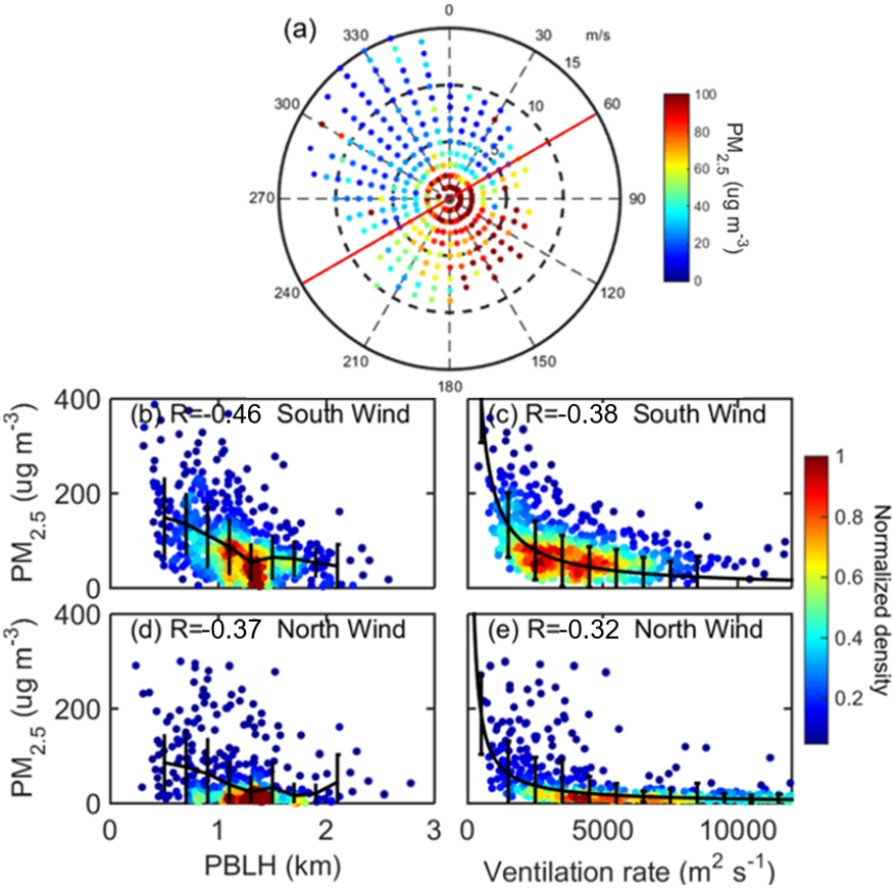


**Figure 7.** (a) Relationship between wind direction/wind speed and PM$_{2.5}$ over Beijing. The red line

divides the northerly wind and southerly wind. (b-c) The relationship between PM$_{2.5}$ and MPL-

PBLH/ventilation rate (VR = WS × PBLH), for southerly winds over Beijing. (d-e) The relationship

between PM$_{2.5}$ and MPL-PBLH/VR, for northerly winds over Beijing.








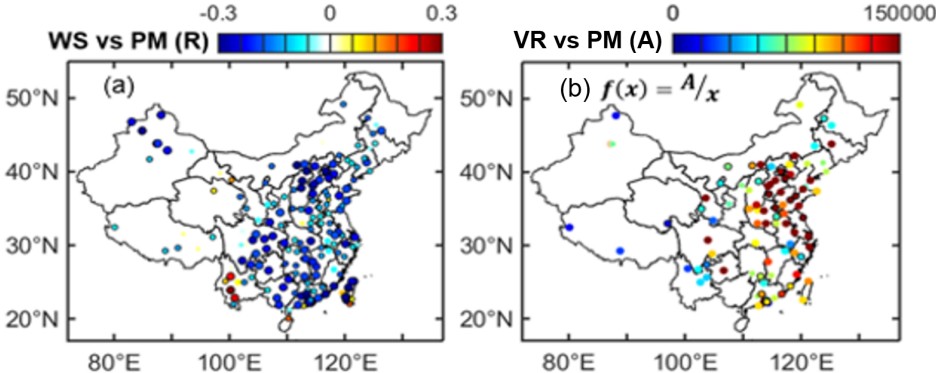


**Figure 8.** (a) Spatial distribution of correlation coefficients (R) for the WS-PM$_{2.5}$ relationship. (b) Spatial

distribution of fitting parameter (A) for the VR-PM$_{2.5}$ relationship. The function $f(x) = {^A}/_x$ is used to

characterize the relationship between VR and PM$_{2.5}$, with A as the fitting parameter, and $x$ is VR, and $f(x)$

is PM$_{2.5}$. Both WS and PM$_{2.5}$ are obtained from surface data, and PBLH are derived from CALIPSO.

Dots marked with black circles indicate where the relationship is statistically significant at the 95%

confidence level.



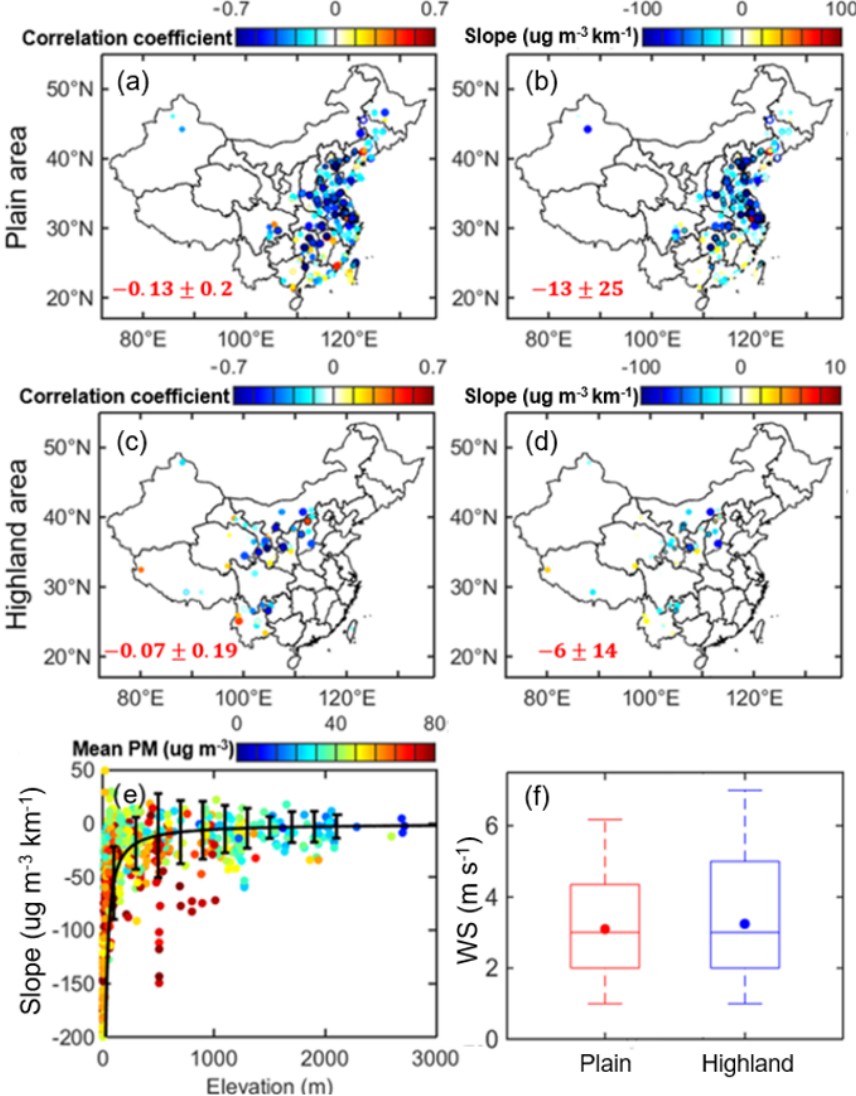

723

**Figure 9.** Stratification by terrain elevation. The correlation coefficients (R) and slopes between

CALIPSO PBLH and noontime $PM_{2.5}$ for plain areas (a-b) and highland areas (c-d). (e) The relationship

between PBLH- $PM_{2.5}$ slope and station elevation, with color-shading indicating station mean $PM_{2.5}$

concentration. (f) Box-and-whisker plots showing the 10th, 25th, 50th, 75th, and 90th percentile values

of the noontime WS for plain and highland regions. The dots indicate the mean value.





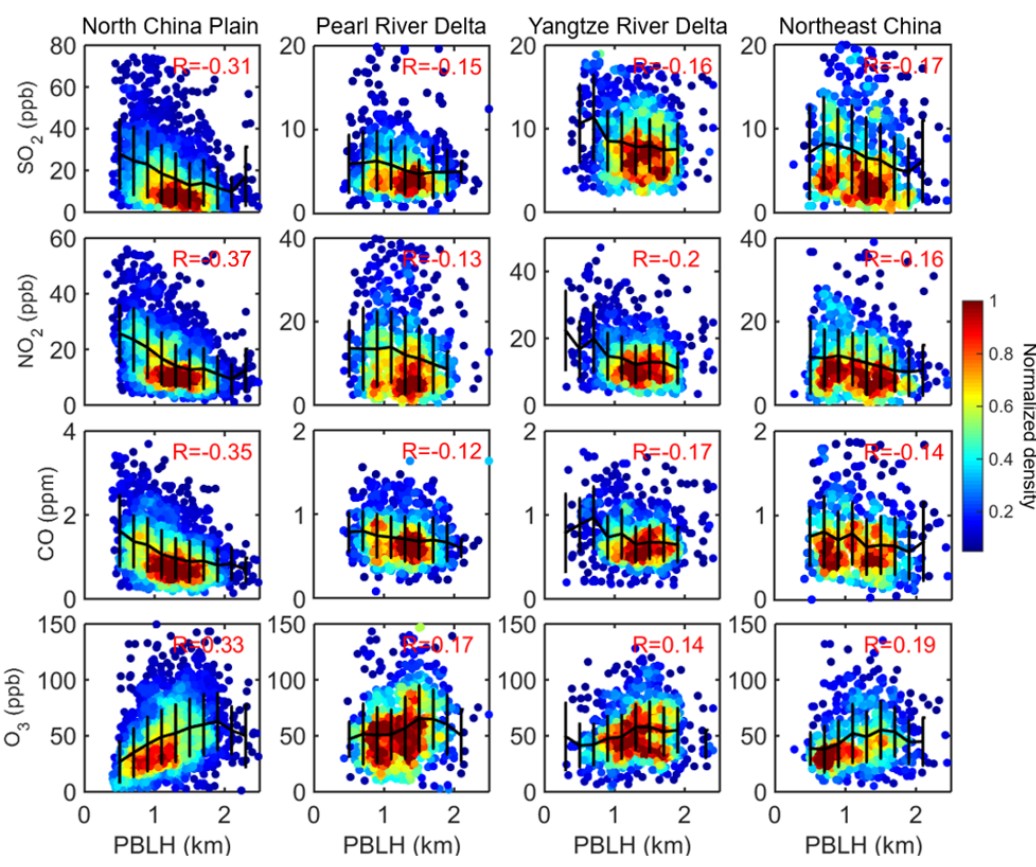

**Figure 10.** The relationships between CALIPSO-derived PBLH and multiple gas pollutants over (from

left to right) the NCP, PRD, YRD, and NEC. The color-shaded dots indicate the normalized sample

density. Correlation coefficients (R) are shown in red in each panel.





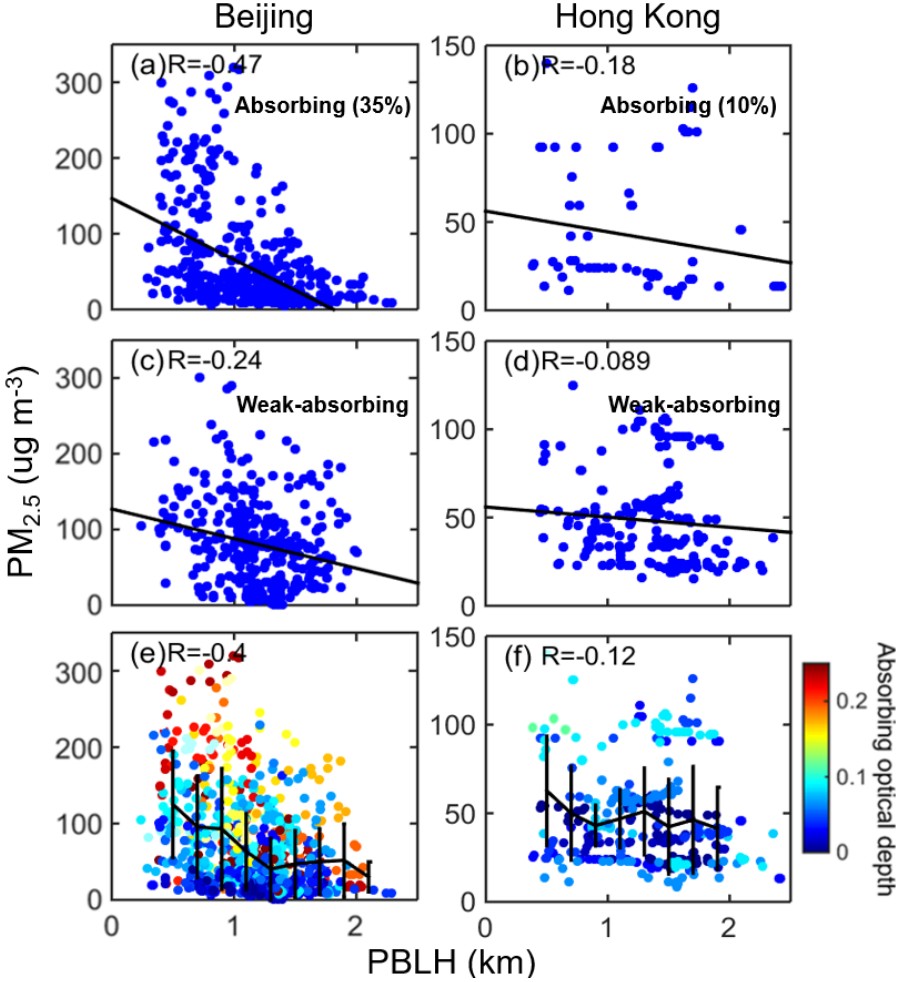


**Figure 11.** The relationship between $PM_{2.5}$ and PBLH for absorbing aerosols over (a) Beijing and (b)


Hong Kong. The percentage of absorbing cases are noted in (a) and (b). The relationship between $PM_{2.5}$


and PBLH for weakly absorbing aerosols over (c) Beijing and (d) Hong Kong. In (a, b, c, d), color bars


represent normalized density. The relationship between total $PM_{2.5}$ and PBLH over (e) Beijing and (f)


Hong Kong. In (e, f), the color-shaded dots indicate absorbing optical depth. The PBLHs over Beijing


are obtained from MPL, and the PBLHs over Hong Kong are calculated by CALIPSO.

