# Peer review of "Relationships between the planetary boundary layer height and"

_Atmospheric Chemistry and Physics, 2018_

## Referee Comment (RC1) · Anonymous Referee #2 · 25 Jun 2018

General Comments: The manuscript studied the relationship between PBLH and PM2.5 concentration over different regions and seasons. Effects of aerosol, winds peed, topography etc. are also included in this study. Many data sources are included, multiple PBLH derived methods are compared, complex statistical relationships are revealed. Thus this study is comprehensive and valuable. While I do have some major revision suggestions since some part of the paper are confusing.

Specific Comments: 1) Section 2 is very confusing. I understand that this part describes many observation datasets including ground based (routine and campaign) and satellite. Also includes multiple PBLH derivation methods. Please reorganize the

section so that readers can have a very clear idea of the data sources and the purpose of the data. Two subsections of 2.1 Data and 2.2 PBLH derive method is good enough. For Data section, use a table to describe all the data used in this study. I included a sample table here. Current section 2.1 is a description of ground based observations, so CALIPSO related statements (line 126-130) are not fit in here. Please move the sentences to section 2.3 PBLH derived from CALIPSO.

2) PBLH is a fundamental variable in this study. Three observational dataset were used to derive PBL: ground MPL, space borne (CALIPSO), and radiosonde. CALIPSO-PBLH is verified by MPL-PBLH, MPL-PLBH is verified by radiosonde-PBLH. These three PBLH derivation methods have different theory bases which contributes discrepancies among them. Statistics as showed in Figure S1 are important, while please give examples of individual comparisons, e.g. one case of PBLH derivations from all the three observations/methods. Another suggestion is to include illustration figures for PBLH determination processes for both MPL and CALIPSO.

3) Section 2.4 MODIS AOD data is suddenly appeared and no explanation of how the data are going to be used and readers have to figure out after read the whole paper. Please add one or two sentences at the beginning to explain the usage. 4) Line 206-210: please move the brief description of MERRA data to Section 2. 5) Reorganize Figure 2 for easy comparison, suggestion: CALISPO at the left column, corresponding MERRA at the second column. 6) Table 1 is very hard to interpret. I suggest to put it in a figure with two y axes, left axis is for PBLH mean and std, right axis for PM2.5. x axis for four regions.

Please also note the supplement to this comment:
https://www.atmos-chem-phys-discuss.net/acp-2018-279/acp-2018-279-RC1-supplement.pdf
* * *
[Figure]

2018.

---

## Referee Comment (RC2) · Anonymous Referee #1 · 29 Jun 2018

This paper investigated the relationship between PBLH and surface PM concentrations over China. The interaction between PBLH and surface pollutants under different topographic and meteorological conditions has been carefully considered. However, I have some concerns about the conclusion of the paper. The authors have investigated many parameters that may influence the relationship between PBLH and surface PM concentrations. But all the derived correlations are relatively low. It seems risky to get the strong conclusion based on those correlations.

General comments:

1. Page 10. I recommend to put the introduction of MERRA in Section 2, which will

make the flow more clear.

2. Section 3.1. The authors discussed the differences between CALIPSO and MERRA in detail. Will the differences influence the conclusion about the relationship between PBLH and surface pollutants? If so, how much will it be?

3. Section 3.2. The correlation coefficient is very low here (Figure 3). I guess it is too risky to make the statement that PBLH has negative correlation with PM2.5 without conditions, which appeared in both abstract and conclusion.

4. The authors use many figures in the supplement to support the discussion in the text. Meanwhile, the text is not self-explanatory without the graphs. I suggest the authors to reconsider the arrangement of the whole graphs including what has been included in the manuscript. You may want to delete the description that is not very relative or add some figures that are really necessary.

5. Section 3.6. I understand the authors would like to perform some preliminary analysis here. But exploring the feedback of absorbing aerosols by only analyzing the correlation between PBLH and PM2.5 looks not convincing for me. You may want to perform some further analysis to make the conclusion more solid or discard this part.

6. I suggest the authors to add the applications of the findings in the conclusion. How will the findings influence the model development or policy design in the future?

Specific comments:

1. Page 3, line 48, the term of "anthropogenic gases" sounds strange. Anthropogenic emissions?

2. Page 3, line 49, "they are much more visible". Please clarify what are compared with.

3. Page 6, line 114. The grammar seems not proper.

4. Page 6, line 124. The source of the meteorological data is missing.

[Figure]

5. Page 6, line 129. The reason for the usage of "noontime" day is missing.

6. Page 11, line 234. The English looks not proper in "The PM2.5 seasonal pattern is generally opposite that of PBLH".

7. Page 11, line 238. The grammar seems not proper.
* * *

---

## Referee Comment (RC3) · Anonymous Referee #3 · 9 Jul 2018

The manuscript investigates relationship between the PBLH and surface PM based on ground-based and onboard lidar, ground environmental and meteorological observations, reanalysis data, and so on. The relationships at different topographic and meteorological conditions over China are specially considered. Although most, if not all, variables show a relatively low correlation with the PBLH, the comprehensive and systematic study reveal the difficulties to drew the relationship between PBLH and surface PM. Generally, the manuscript discusses an important topic, and the methods and discussions are solid and meaningful.

General comments:

[Figure]

1. Some general information about the environmental and meteorological stations used for the four regions should be presented, such as number of stations used in each region, the basic types of them (are them all in the city?). Is there any quality control carried out for the results?

2. Figure 2 can be reorganized for better comparison. The CALIPSO and MERRA results can be shown in the left and right panel, respectively, and, then, results from the same season can be directly compared.

3. The MERRA PBLH is not well introduced in the text. Meanwhile, after Figure 2, most results are compared with the CALIPSO results. The MERRA data can be used to evaluate the CALIPSO data, and if it is not used in the discussion for relationship with the PM, why the authors still discuss it in the manuscript.

4. Section 3.5 and Figure 10 that show the relationship between multiple gases and PBHL are the only part discussing about the gases. Again, relatively poor corrections are obtained, and also considering that this study focuses on the relationship of PBHL and PM, it is not necessary to present those results. This will keep the manuscript more focused.

5. Even the relationship between PM and PBLH is relatively weak, how would it possible to further discuss the aerosol absorption feedback in section 3.6.

6. Considering the relatively low correlations shown in the paper, the conclusions are too strong. For example, in the abstract, the authors mentioned that "(line 31) A generally negative correlation is obtained between PM and the PBLH", while the largest correction obtained is only 0.36 from Figure 3. Multiple 'strong correlations' are mentioned in conclusion section.

7. Besides the conclusions, some relatively strong statements in the manuscript should be reconsidered. For example, on line 146, "This method can handle all possible weather conditions and aerosol layers. . . . . .."

---

## Author Comment (AC1) · 9 Sep 2018

*Response to Reviewer #1:*

*This paper investigated the relationship between PBLH and surface PM concentrations over China. The interaction between PBLH and surface pollutants under different topographic and meteorological conditions has been carefully considered. However, I have some concerns about the conclusion of the paper. The authors have investigated many parameters that may influence the relationship between PBLH and surface PM concentrations. But all the derived correlations are relatively low. It seems risky to get the strong conclusion based on those correlations.*

**Response: We are very grateful to the reviewer for his/her valuable comments on our work, which are quite constructive and helpful. We carefully considered all of these comments, and modified some strong conclusion regarding the PBLH-PM relationships, as well as the analysis. Our detailed responses to the reviewer's questions and comments are listed below.**

*General comments:*

*1. Page 10. I recommend to put the introduction of MERRA in Section 2, which will make the flow more clear.*

**Response: Per your kind suggest, we moved the introduction of MERRA data to Section 2 (revised Section 2.2.3).**

*2. Section 3.1. The authors discussed the differences between CALIPSO and MERRA in detail. Will the differences influence the conclusion about the relationship between PBLH and surface pollutants? If so, how much will it be?*

**Response: Thanks for this valuable comment. In fact, the reanalysis data take account of large-scale dynamical forcing, and have the ability to produce a general PBLH climatology (Guo et al., 2016) which is used to compare with that derived from CALIPSO in this study. However, the reanalysis data do not consider the impact of aerosols; only limited upper atmospheric measurements are assimilated, and the effects of aerosol-PBL interaction are poorly represented (Ding et al., 2013; Simmons, 2006; Huang et al., 2018). Thus, the reanalysis data offer limited ability to investigate detailed PBLH-PM relationships. Therefore, the observation-based retrievals (CALIPSO PBLH or MPL PBLH) are used to produce the PBLH-PM relationships over China. A detailed discussion has been incorporated into the revised Section 3.1.**

*3. Section 3.2. The correlation coefficient is very low here (Figure 3). I guess it is too risky to make the statement that PBLH has negative correlation with PM2.5 without conditions, which appeared in both abstract and conclusion.*

**Response: We greatly appreciate this constructive comment. Indeed, the PBLH is not always negatively correlated with $PM_{2.5}$. The weak correlation coefficients cause some**

difficulties in deriving a clear relationship between PBLH and $PM_{2.5}$. In addition to PBLH, $PM_{2.5}$ is also controlled by many other factors (e.g. emissions, wind, synoptic patterns, stability, etc.), and thus, the variation of $PM_{2.5}$ is not necessarily related to PBLH, especially when other factors play dominant roles (e.g. strong wind). In such situations, there are rather weak or uncorrelated relationships between PBLH and $PM_{2.5}$. Strong aerosol-PBL interactions only occur under certain conditions. In our analysis, heavy aerosol loading, plains areas, and weak wind speed would be favorable conditions for relatively strong negative correlations between PBLH and $PM_{2.5}$. This discussion has been incorporated into the revised Section 4.

In addition, we revised the overly strong statements to avoid misleading the reader, and show three examples as follows:

In the abstract, "A generally negative correlation is observed between PM and the PBLH…" has been revised to "Albeit the PBLH-PM correlations are roughly negative for most cases, their magnitude, significance, and even sign vary considerably with location, season, and meteorological conditions."

In conclusion, "We observe widespread negative correlations…" has been revised to "Albeit the PBLH-$PM_{2.5}$ correlations are generally negative for the majority of conditions, their magnitude, significance, and even sign vary greatly by region and timing."

In conclusion, "Strong correlations between PBLHs and aerosols occur in low-altitude regions." has been revised to "The PBLH-$PM_{2.5}$ correlations are found to be more significant in low-altitude regions."

Moreover, we previously used the Pearson correlation coefficient, which is representative in a linear relationship. However, the PBLH-$PM_{2.5}$ relationships are nonlinear under most conditions, and this fact would contribute to the low Pearson correlation coefficients. To partly address this problem, we introduce an inverse function ($f(x) = A/x + B$) to fit the PBLH-$PM_{2.5}$ relationships more closely with set the weighting function as the normalized density. In Figure R1 (the revised Figure 5), we jointly use the regular linear regression and the fitted inverse function to characterize the PBLH-$PM_{2.5}$ relationships. Over North China Plain, the nonlinear inverse function shows high consistency with the average values for each bin, and well represents the behavior of the most dense area in the scatter plot with an improved correlation (correlation coefficient -0.49). Similar improvements in the fitting method are also found in other regions, but are still not significant for Pearl River Delta and Northeast China (relatively clean regions).

The fitting methods are described in the revised Section 2.3. We updated the fitting method description in the revised manuscript, which shows better performance in characterizing the PBLH-$PM_{2.5}$ relationships.

[Figure]

**Figure R1.** The relationship between CALIPSO-derived PBLH and early-afternoon PM$_{2.5}$ over (a) NCP, (b) PRD, (c) YRD, and (d) NEC. The black dots and whiskers represent the average values and standard deviation for each bin. The red dash lines indicate the regular linear regressions, and the black lines represent the inverse fit ($f(x) = \frac{A}{x} + B$). The detailed fitting functions are given at the top of each panels, along with the Pearson correlation coefficient (red) and the correlation coefficient for the inverse fit (black). Here and in the following analysis, R with asterisks indicates the correlation is statistically significant at the 99% confidence level. The color-shaded dots indicate the normalized sample density.

*4. The authors use many figures in the supplement to support the discussion in the text. Meanwhile, the text is not self-explanatory without the graphs. I suggest the authors to reconsider the arrangement of the whole graphs including what has been included in the manuscript. You may want to delete the description that is not very relative or add some figures that are really necessary.*

**Response: Thanks for pointing this out. We reorganized the supporting information (SI). Figure S2 was revised to present the PM$_{2.5}$ climatology, and was moved to the main text (the revised Figure 4). Figure S4 was revised to present the relationship between MPL PBLH and PM$_{2.5}$ (or normalized PM$_{2.5}$) at Beijing, and was moved to the main text (the revised Figure 7). Previous Figure S3 and Figure S9 were deleted from the SI. As such, we**

believe that our main points are delivered and reflected in the revised main text. The SI presents some additional analyses to complement the points in the main text with more evidences.

*5. Section 3.6. I understand the authors would like to perform some preliminary analysis here. But exploring the feedback of absorbing aerosols by only analyzing the correlation between PBLH and PM2.5 looks not convincing for me. You may want to perform some further analysis to make the conclusion more solid or discard this part.*

Response: We appreciate your suggestion. Indeed, the analysis in Section 3.6 is insufficient, and we deleted this section as suggested. In the discussion, we mention the feedback of absorbing aerosols would be a potential factor affecting the PBLH-PM relationships, which merits further analysis that will be presented in a future paper, given the long length of the current paper.

*6. I suggest the authors to add the applications of the findings in the conclusion. How will the findings influence the model development or policy design in the future?*

Response: Per your comment, we added following statement to the Section 4:

"Such information can help improve our understanding of the complex interactions between air pollution, boundary layer, and horizontal transport, and thus, can benefit policy making aimed at mitigating air pollution at both local and regional scales. Our study also contributes to the quantitative understanding of aerosol-PBL interaction and further improvement of surface pollutant monitoring and forecasting capabilities."

*Specific comments:*

1. *Page 3, line 48, the term of "anthropogenic gases" sounds strange. Anthropogenic emissions?*

Response: We revised the "anthropogenic gases" to "gaseous pollutants".

2. *Page 3, line 49, "they are much more visible". Please clarify what are compared with.*

Response: Per your comment, we revised the statement as "PM pollutants are of greater concern to the public partly because they are much more visible than gaseous pollution…"

3. *Page 6, line 114. The grammar seems not proper.*

Response: Per your kind suggestion, we revised the statement as "These empirical relationships between PBLH and surface pollutants are expected to improve our understanding and forecast capability for air pollution..."

4. *Page 6, line 124. The source of the meteorological data is missing.*

Response: Per your comment, we added the source.

5. *Page 6, line 129. The reason for the usage of "noontime" day is missing.*

**Response: Thanks for pointing this out. We changed "noontime" to "early-afternoon" and added some clarifying text:**

**"To match the CALIPSO retrievals with equator crossings at approximately 1330 local time, we use the surface meteorological and environmental data in early-afternoon, averaged from 1300 to 1500 China standard time (CST). During this period, the PBL is well developed with relatively strong vertical mixing, which is a favorable condition for investigating aerosol-PBL interaction."**

*6. Page 11, line 234. The English looks not proper in "The PM2.5 seasonal pattern is generally opposite that of PBLH".*

**Response: We revised the statement as "The PM$_{2.5}$ seasonal pattern is generally coupled to that of PBLH…"**

*7. Page 11, line 238. The grammar seems not proper.*

**Response: We revised the statement as "Both the PBLH and PM$_{2.5}$ also show strong seasonality over NCP. PRD is a relatively clean region, and PM$_{2.5}$ maintains low values (<50 µg m$^{-3}$) through all seasons"**

**References:**

Guo, J., Miao, Y., Zhang, Y., Liu, H., Li, Z., Zhang, W., He, J., Lou, M., Yan, Y., Bian, L. and Zhai, P.: The climatology of planetary boundary layer height in China derived from radiosonde and reanalysis data. Atmos. Chem. Phys., 16(20), 13,309–13,319. https://doi.org/10.5194/acp-16-13309-2016, 2016.

Huang, X., Wang, Z. and Ding, A.: Impact of Aerosol-PBL Interaction on Haze Pollution: Multi-Year Observational Evidences in North China. Geophysical Research Letters, 2018.

Ding, A. J., et al., Intense atmospheric pollution modifies weather: a case of mixed biomass burning with fossil fuel combustion pollution in eastern China, Atmos. Chem. Phys., 13(20), 10545-10554, 2013.

Simmons, A., ERA-Interim: New ECMWF reanalysis products from 1989 onwards, ECMWF newsletter, 110, 25-36., 2006.

---

## Author Comment (AC2) · 9 Sep 2018

**Response to Reviewer #2:**

***General Comments:***

*The manuscript studied the relationship between PBLH and PM2.5 concentration over different regions and seasons. Effects of aerosol, winds peed, topography etc. are also included in this study. Many data sources are included, multiple PBLH derived methods are compared, complex statistical relationships are revealed. Thus this study is comprehensive and valuable. While I do have some major revision suggestions since some part of the paper are confusing.*

**Response: We are very grateful to the reviewer for his/her helpful and constructive comments on our work. All of the comments and concerns raised by the referee have been carefully considered and incorporated into this revision. Our detailed responses to the reviewer's questions and comments are listed below.**

***Specific Comments:***

*1) Section 2 is very confusing. I understand that this part describes many observation datasets including ground based (routine and campaign) and satellite. Also includes multiple PBLH derivation methods. Please reorganize the section so that readers can have a very clear idea of the data sources and the purpose of the data. Two subsections of 2.1 Data and 2.2 PBLH derive method is good enough. For Data section, use a table to describe all the data used in this study. I included a sample table here. Current section 2.1 is a description of ground based observations, so CALIPSO related statements (line 126-130) are not fit in here. Please move the sentences to section 2.3 PBLH derived from CALIPSO.*

**Response: Thanks a lot for the guidance. Following your instruction, we added Table R1 to section 2 to describe the observations from multiple sources and platforms.**
**Table R1. Description of data.**

| Observations | Variables | Location | Temporal resolution | Time period |
|---|---|---|---|---|
| Environmental Stations | PM$_{2.5}$ | ~1600 sites* | Hourly | 01/2012-06/2017 |
| Meteorological Stations | WS/WD | ~900 sites** | Hourly | 01/2012-06/2017 |
| MPL | PBLH, extinction | Beijing | 15seconds | 03/2016-12/2017 |
| AERONET | AOD (550nm), | Beijing | ~Hourly | 01/2016-12/2017 |
| MODIS | AOD | Whole China | Daily | 01/2006-12/2017 |
| CALIPSO | PBLH | Orbits in Figure 1d | Daily | 06/2006-12/2017 |
| MERRA | PBLH | Whole China | Hourly | 01/2006-12/2017 |

\* 224 sites over NCP; 105 sites over PRD; 215 sites over YRD; 159 sites over NEC

\*\* 37 sites over NCP; 92 sites over PRD; 34 sites over YRD; 76 sites over NEC

**In addition, we reorganized section 2, and kept two subsections describing the data and PBLH methodology, respectively. We also added a subsection to illustrate the statistical analysis methods. The CALIPSO-related statements in section 2.1 have been moved to the revised section 2.1.2.**

*2) PBLH is a fundamental variable in this study. Three observational dataset were used to derive PBL: ground MPL, space borne (CALIPSO), and radiosonde. CALIPSO-PBLH is verified by MPL-PBLH, MPL-PLBH is verified by radiosonde-PBLH. These three PBLH derivation methods have different theory bases which contributes discrepancies among them. Statistics as showed in Figure S1 are important, while please give examples of individual comparisons, e.g. one case of PBLH derivations from all the three observations/methods. Another suggestion is to include illustration figures for PBLH determination processes for both MPL and CALIPSO.*

**Response: Per your sound suggestion, we updated the statistical analysis for the PBLH comparisons. The RMSE and sample numbers (N) are given in each panel, the correlation coefficients (R) are already given in each panel, and R with asterisks indicates those correlations that are statistically significant above the 99% confidence level. As an example. Figure R1 (the**

**revised Figure S1) show the PBLH retrievals derived from CALIPSO, MPL, and RS on 7 June 2016 over Beijing. Based on aerosol backscatter, CALIPSO and MPL derive consistent PBLH retrievals in this case. Radiosonde also show reasonably good agreement with CALIPSO and MPL retrievals with a difference of ~0.1km.**

[Figure]

**Figure R1.** (a) Time evolution of the normalized signal (NS) plot from MPL on 7 June 2016 over Beijing. The black dots identify the PBLH derived from MPL, and the blue star indicates the PBLH derived from radiosonde. (b) Total attenuated backscatter (TAB) plot (log scale) from CALIPSO on 7 June 2016 over Beijing. The black line indicate the PBLH derived from CALIPSO. The red dot represents the corresponding PBLH derived from MPL, and the blue star indicates the PBLH derived from radiosonde.

    **As CALIPSO provides the primary measurements used in this study, we added a figure illustrating the CALIPSO PBLH determination processes (the revised Figure 2). For retrieving PBLH from MPL, we implement a well-established method, which was developed by Yang et al. (2013) and was adopted in multiple studies (e.g. Lin et al., 2016; Su et al., 2017). The principle is based on the traditional gradient method, and people can access the published paper (Yang et al., 2013) if they seek more details. We might save some space if we do not add the illustration figure for MPL.**

[Figure]

**Figure R2.** The schematic diagram of retrieving the PBLH from CALIPSO.

*3) Section 2.4 MODIS AOD data is suddenly appeared and no explanation of how the data are going to be used and readers have to figure out after read the whole paper. Please add one or two sentences at the beginning to explain the usage.*

**Response: Thanks for pointing this out; we added the following statements to Section 2 to explain the use of MODIS AOD data:**

**"Note that aerosol loading is significantly different in different regions. To account for the background pollution level, we normalize the $PM_{2.5}$ with the MODIS AOD to qualitatively account for background or transported aerosol that is not concentrated in the PBL."**

*4) Line 206-210: please move the brief description of MERRA data to Section 2.*

**Response: According to your comment, we moved the description of the MERRA data to Section 2 as a new subsection (revised section 2.2.3).**

*5) Reorganize Figure 2 for easy comparison, suggestion: CALISPO at the left column, corresponding MERRA at the second column.*

**Response: Per your suggestion, we have revised this figure.**

*6) Table 1 is very hard to interpret. I suggest to put it in a figure with two y axes, left axis is for PBLH mean and std, right axis for PM2.5. x axis for four regions.*

**Response: Thanks for this valuable comment. The table is indeed very hard to interpret and conveys little scientific value, and is thus, deleted from the main text but keep as supporting information in case of anyone interested knowing the values of PBLH and $PM_{2.5}$ over different**

**ROIs. We revised Figure 3 and Figure 4 showing the climatological patterns of PBLH and PM$_{2.5}$ that are visually revealing. `**

**References:**
Su, T., Li, J., Li, C., Xiang, P., Lau, A.K.H., Guo, J., Yang, D., and Miao, Y.: An inter-comparison of long-term planetary boundary layer heights retrieved from CALIPSO, ground-based lidar, and radiosonde measurements over Hong Kong. J. Geophys. Res., 122(7), pp.3929-3943, 2017.

Lin, C.Q., Li, C.C., Lau, A.K., Yuan, Z.B., Lu, X.C., Tse, K.T., Fung, J.C., Li, Y., Yao, T., Su, L. and Li, Z.Y.: Assessment of satellite-based aerosol optical depth using continuous lidar observation. Atmospheric environment, 140, pp.273-282, 2016.

Yang, D., Li, C., Lau, A. K. H., and Li, Y.: Long-term measurement of daytime atmospheric mixing layer height over Hong Kong. J. Geophys. Res., 118, 2,422–2,433. https://doi.org/10.1002/jgrd.50251, 2013.

---

## Author Comment (AC3) · 9 Sep 2018

**Response to Reviewer #3:**

*The manuscript investigates relationship between the PBLH and surface PM based on ground-based and onboard lidar, ground environmental and meteorological observations, reanalysis data, and so on. The relationships at different topographic and meteorological conditions over China are specially considered. Although most, if not all, variables show a relatively low correlation with the PBLH, the comprehensive and systematic study reveal the difficulties to drew the relationship between PBLH and surface PM. Generally, the manuscript discusses an important topic, and the methods and discussions are solid and meaningful.*

**Response: We are very grateful to the reviewer for his/her valuable and constructive comments on our work. All of these comments and concerns raised by the referee have been carefully considered and incorporated into this revision. Our detailed responses to the reviewer's questions and comments are listed below.**

*General Comments:*

*1. Some general information about the environmental and meteorological stations used for the four regions should be presented, such as number of stations used in each region, the basic types of them (are them all in the city?). Is there any quality control carried out for the results?*

**Response: Thanks for the valuable suggestion. We added Table R1 to section 2 to summarize the data. Table R1 not only reports the number of meteorological and environmental stations in each region, but also gives general information about the data used from other sources. The station locations are not all in the cities, but are widely distributed in both urban and rural areas. However, in this large-scale study, we stratify by geographic region, and do not consider the differences between the rural and urban areas specifically.**

**Table R1. Description of data.**

| Observations | Variables | Location | Temporal resolution | Time period |
|---|---|---|---|---|
| Environmental Stations | $PM_{2.5}$ | ~1600 sites* | Hourly | 01/2012-06/2017 |
| Meteorological Stations | WS/WD | ~900 sites** | Hourly | 01/2012-06/2017 |
| MPL | PBLH, extinction | Beijing | 15seconds | 03/2016-12/2017 |
| AERONET | AOD (550nm), | Beijing | ~Hourly | 01/2016-12/2017 |
| MODIS | AOD | Whole China | Daily | 01/2006-12/2017 |
| CALIPSO | PBLH | Orbits in Figure 1d | Daily | 06/2006-12/2017 |
| MERRA | PBLH | Whole China | Hourly | 01/2006-12/2017 |

\* 224 sites over NCP; 105 sites over PRD; 215 sites over YRD; 159 sites over NEC

\*\* 37 sites over NCP; 92 sites over PRD; 34 sites over YRD; 76 sites over NEC

**These meteorological and environmental data are routinely measured and quality controlled by government agencies. The $PM_{2.5}$ dataset has been evaluate by other study, and shows relatively high reliability (Liang et al., 2016). There are quality flags along with the meteorological measurements, so error data can be eliminated. These points have been incorporated into the revised Section 3.1.**

*2. Figure 2 can be reorganized for better comparison. The CALIPSO and MERRA results can be shown in the left and right panel, respectively, and, then, results from the same season can be directly compared.*

**Response: Per your kind comment, we revised this figure.**

*3. The MERRA PBLH is not well introduced in the text. Meanwhile, after Figure 2, most results are compared with the CALIPSO results. The MERRA data can be used to evaluate the CALIPSO data, and if they are not used in the discussion for relationship with the PM, why the authors still discuss it in the manuscript.*

**Response: Thanks for pointing this out. We have added a brief introduction to the MERRA data in Section 2.2.3. As the reanalysis data take account of large-scale dynamic forcing, they are used to produce the climatology pattern of PBLH, and compared with those derived from CALIPSO. We found that the CALIPSO and MERRA retrievals exhibit some mutual features in the seasonality, which is roughly coupled with the seasonal climatology of PM$_{2.5}$. However, we do not focus on the detailed MERRA PBLH values, so we removed the original Table 1 in the main text.**

**In fact, the reanalysis data bear the model uncertainties, and do not include the impact of aerosols except based on the limited upper atmospheric measurements assimilated (Simmons, 2006). As results, these data poorly represent the effects of aerosol-PBL interactions (Ding et al., 2013; Huang et al., 2018), and offer limited ability to investigate detailed PBLH-PM relationships. As a result, we use only the observation-based retrievals (CALIPSO PBLH or MPL PBLH) to produce the PBLH-PM relationships over China. This discussion has been incorporated into the revised Section 3.1.**

*4. Section 3.5 and Figure 10 that show the relationship between multiple gases and PBHL are the only part discussing about the gases. Again, relatively poor corrections are obtained, and also considering that this study focuses on the relationship of PBHL and PM, it is not necessary to present those results. This will keep the manuscript more focused.*

**Response: Per your kind guidance, we deleted this section and Figure 10.**

*5. Even the relationship between PM and PBLH is relatively weak, how would it possible to further discuss the aerosol absorption feedback in section 3.6.*

**Response: We deleted this section as suggested, and only mention that the feedback of absorbing aerosols could be a potential influencing factor that merits further analysis.**

*6. Considering the relatively low correlations shown in the paper, the conclusions are too strong. For example, in the abstract, the authors mentioned that "(line 31) A generally negative*

*correlation is obtained between PM and the PBLH", while the largest correction obtained is only 0.36 from Figure 3. Multiple 'strong correlations' are mentioned in conclusion section.*

**Response: We appreciate your kind suggestion. Indeed, since PM$_{2.5}$ is controlled by many other factors (e.g. emission, wind, synoptic pattern, stability, etc.), the correlations between PBLH and PM$_{2.5}$ are not very strong under most conditions. We revised the statements in conclusions section to avoid overly strong statements, and state that "Albeit the PBLH-PM$_{2.5}$ correlations are generally negative for the majority conditions, their magnitude, significance, and even sign vary greatly with location, season, and meteorological conditions". We also emphasize that relatively strong PBLH-PM$_{2.5}$ correlations only occurred under certain conditions. According to our analysis, heavy aerosol loading, the plains area, and weak wind speed would be favorable conditions for relatively strong negative correlations between PBLH and PM$_{2.5}$. These points have been incorporated into the revised Section 4.**

**Moreover, we previously used the Pearson correlation coefficient derived from the linear relationship. However, the PBLH-PM$_{2.5}$ relationships are nonlinear under most conditions as shown in the figure here. Thus, the nonlinear relationships would contribute to the low Pearson correlation coefficients. To partly address this problem, we included a new fitting method based on an inverse function ($f(x) = A/x + B$) to characterize the PBLH-PM$_{2.5}$ relationships, and set the weighting function as the normalized density. As shown in Figure R1 (the revised Figure 5), the nonlinear inverse function fits show better performance with the data, and characterize the behavior of the most dense area in the scatter plot with improved correlation coefficient (-0.49). Therefore, we include the new fitting method in the revised manuscript, which shows better performance in characterizing the PBLH-PM$_{2.5}$ relationships.**

[Figure]

**Figure R1.** The relationship between CALIPSO-derived PBLH and early-afternoon PM$_{2.5}$ over

(a) NCP, (b) PRD, (c) YRD, and (d) NEC. The black dots and whiskers represent the average values and standard deviation for each bin. The red dash lines indicate the regular linear regressions, and the black lines represent the inverse fit ($f(x) = A/x + B$). The detailed fitting functions are given at the top of each panels, along with the Pearson correlation coefficient (red) and the correlation coefficient for the inverse fit (black). Here and in the following analysis, R with asterisks indicates the correlation is statistically significant at the 99% confidence level. The color-shaded dots indicate the normalized sample density.

*7. Besides the conclusions, some relatively strong statements in the manuscript should be reconsidered. For example, on line 146, "This method can handle all possible weather conditions and aerosol layers. . .. . .."*

**Response: Per your kind suggestion, we checked the manuscript and revised or delete these improper statements.**

**References:**

Huang, X., Wang, Z. and Ding, A.: Impact of Aerosol-PBL Interaction on Haze Pollution: Multi-Year Observational Evidences in North China. Geophysical Research Letters, 2018.

Ding, A. J., et al., Intense atmospheric pollution modifies weather: a case of mixed biomass burning with fossil fuel combustion pollution in eastern China, Atmos. Chem. Phys., 13(20), 10545-10554, 2013.

Simmons, A., ERA-Interim: New ECMWF reanalysis products from 1989 onwards, ECMWF newsletter, 110, 25-36., 2006.

Liang, X., S. Li, S. Y. Zhang, H. Huang, and S. X. Chen (2016), PM2.5 data reliability, consistency, and air quality assessment in five Chinese cities, J Geophys Res-Atmos, 121(17), 10220-10236.

---

## Referee Report (RR1)

The manuscript has been greatly improved and logically clear. All the dataset used in the study have been clearly and properly descripted, all the methods are introduced and illustrated. Results are sound and conclusions are appropriately follow the results.

I recommend acceptance. There is only one minor I noticed:

Line 556, reference Rienecker et al. 2011 is not listed at its corresponding alphabetic place.